# 3D-AWARE DISENTANGLED REPRESENTATION FOR COMPOSITIONAL REINFORCEMENT LEARNING

[1]**Sungbin Mun**, [1]**Younghwan Lee**, [2]**Cheul-Hui Min**, [3]**Mineui Hong**, [1]**Young Min Kim**
[1]Seoul National University   [2]Samsung Electronics   [3]Carnegie Mellon University
{brian0429, frredy99, youngmin.kim}@snu.ac.kr
cheolhui.min@samsung.com   mineuih@andrew.cmu.edu

## ABSTRACT

Vision-based reinforcement learning can benefit from object-centric scene representation, which factorizes the visual observation into individual objects and their attributes, such as color, shape, size, and position. While such object-centric representations can extract components that generalize well for various multi-object manipulation tasks, they are prone to issues with occlusions and 3D ambiguity of object properties due to their reliance on single-view 2D image features. Furthermore, the entanglement between object configurations and camera poses complicates the object-centric disentanglement in 3D, leading to poor 3D reasoning by the agent in vision-based reinforcement learning applications. To address the lack of 3D awareness and the object-camera entanglement problem, we propose an enhanced 3D object-centric representation that utilizes multi-view 3D features and enforces more explicit 3D-aware disentanglement. The enhancement is based on the integration of the recent success of multi-view Transformer and the prototypical representation learning among the object-centric representations. The representation, therefore, can stably identify proxies of 3D positions of individual objects along with their semantic and physical properties, exhibiting excellent interpretability and controllability. Then, our proposed block transformer policy effectively performs novel tasks by assembling desired properties adaptive to the new goal states, even when provided with unseen viewpoints at test time. We demonstrate that our 3D-aware block representation is scalable to compose diverse novel scenes and enjoys superior performance in out-of-distribution tasks with multi-object manipulations under both seen and unseen viewpoints compared to existing methods.

## 1 INTRODUCTION

Vision-based reinforcement learning utilizes images as an input modality, enabling a flexible, human-like perception for robotic tasks, compared to specific sensory or motor configurations. However, the unstructured nature of raw observations further challenges the inherent inefficiency in reinforcement learning (RL). As the visual observations from various viewpoints result from a complex interplay between color, lighting, geometry and their non-linear projection, the images do not easily map to a coherent composition of meaningful attributes, which challenges their generalizability and controllability. While a visual perception module can benefit from recent advancements in computer vision literature, the feature-level abstractions are often high-dimensional and are no longer interpretable by humans.

Systematic factorization of visual observation can lead to generalizable perceptual reasoning, one of the desired benefits of vision-based RL (Yoon et al., 2023). Object-centric representations promote component-wise analysis, where individual objects are detached from the background. Furthermore, learning factor-level representations like Neural Systematic Binder (Singh et al., 2022) deduces the spatial locations and visual attributes, such as colors and shapes, from which humans can interpret without additional training. Thus, there have been studies that exploit object-centric representation as a state representation for agent control (Driess et al., 2023; Haramati et al., 2024; Min & Kim, 2025), which has been shown to improve both the task solving performance of vision-based RL agents and the efficiency of the training sample.

However, there remain two key challenges that prevent the agent from fully utilizing object-centric representations: 1) insufficient 3D-awareness and 2) imprecise object description. First, the insufficient 3D-awareness stems from missing multi-view attention and 2D UV-grid-based decoding, and in practice many slot-attention based methods operate on a single 2D view or feature grid, so they cannot reliably infer depth, occlusion, multi-view consistency, or full 3D spatial object poses (Chen et al., 2021; Biza et al., 2023; Min & Kim, 2025). This limitation reduces their applicability in interactive environments with occlusions and viewpoint changes. Second, due to the unsupervised learning process, the object variables remain ambiguous in terms of the agent's manipulability, because slot representations often correspond to clusters in 2D feature maps (e.g., color, texture, or region) and do not guarantee that each slot encodes a distinct physically manipulable object, its pose, its affordances, or its dynamic interactions (Locatello et al., 2020; Kim et al., 2023; Kori et al., 2024; Rezazadeh et al., 2024).

To address these challenges, we propose a novel end-to-end neural network architecture called 3D block-slot representation, providing a 3D-aware and interpretable state estimation to facilitate vision-based RL tasks. Specifically, we adapt the *block-slot* concept from the attribute-level structural factorization (Singh et al., 2022) to facilitate explicit yet unsupervised object-centric decomposition. Further, to achieve 3D-awareness, we lift the decomposed attributes into 3D space using a light-field decoder (Sajjadi et al., 2022a;b; Smith et al., 2022; Yu et al., 2021; Stelzner et al., 2021; Qi et al., 2023; Luo et al., 2024; Liu et al., 2024), enabling camera view-object 3D position disentanglement. Building on our proposed representation, we introduce a block transformer policy that significantly improves the agent's generalization to novel object configurations. By recognizing matching properties between the current and goal states, the policy effectively reduces the search space for goal-conditioned planning.

To sum up, our main contributions are as follows:

- We propose a novel 3D block-slot attention mechanism that enables the construction of view-independent 3D structured representations by disentangling object attributes such as shape, color, size, and position.

- In the 3D block-slot attention, we design a novel slot attention scheme that applies block-slot attention to object slots, while using vanilla slot attention for background and agent slots, enabling consistent and disentangled representations across different scene components.

- We develop a block transformer policy that performs block-wise cross-attention between current and goal states, enabling precise goal conditioning on object attributes.

- Our experiments on attribute-level object manipulation confirm that the 3D block-slot representation is effective for structured and generalizable goal-conditioned RL, providing 3D-aware features that make the policy view-agnostic and robust to unseen viewpoints.

## 2 METHOD

We extract a 3D-aware representation that is decomposed into interpretable components with blocks, such that it can efficiently generalize to various goal-driven tasks with vision-based RL. We first describe the encoder architecture with disentangled components (Section 2.1) followed by the transformer policy architecture (Section 2.2). The overall architecture is summarized in Figure 1.

### 2.1 3D BLOCK-SLOT REPRESENTATION

Our latent embedding encodes a consistent, disentangled representation that can match visual observations in an arbitrary viewpoint, thus being robust to occlusion or perspective distortion. We build upon the 3D object scene representation transformer (OSRT) (Sajjadi et al., 2022a), which disentangles individual instances and further identifies the active and passive components in fulfilling robotic tasks. The additional block-slot attention module (Singh et al., 2022) builds a strong structure with decomposed factors that are associated with various attributes of the objects.

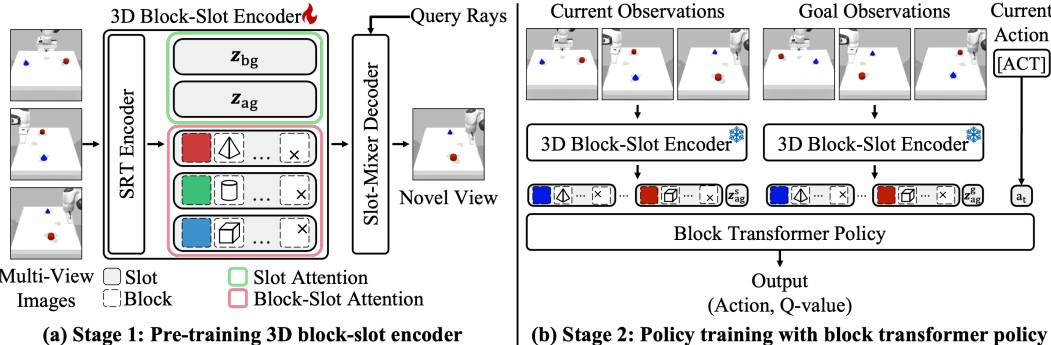

(a) Stage 1: Pre-training 3D block-slot encoder | (b) Stage 2: Policy training with block transformer policy

Figure 1: **Overall structure of our method:** Our proposed pipeline consists of two steps: representation learning and policy training. **(a) Pre-training 3D block-slot encoder**: The object slots are further decomposed into blocks of attributes. Then, the slot-mixer decoder mixes the object-centric representation to generate images at a query view. **(b) Policy training with block transformer policy**: We utilize the 3D block-slot encoder to extract a structured representation for the current observation and the goal image. The decomposed latent embedding serves as the input and the goal tokens, respectively, for our block transformer of the policy architecture.

### 2.1.1 PRELIMINARY: 3D OBJECT-CENTRIC LEARNING

We adopt a transformer-based SRT encoder ($E_\theta$) (Sajjadi et al., 2022b) to aggregate multi-view observations $\{\mathbf{I}_i \in \mathbb{R}^{H \times W \times 3}\}$ and generate a scene-level latent representation $\mathbf{F} \in \mathbb{R}^{L \times D}$:

$$\mathbf{F} = E_\theta(\{\mathbf{I}_i\}). \tag{1}$$

The slot attention mechanism (Locatello et al., 2020) is applied to the scene feature $\mathbf{F}$, producing a set of slots $\{\mathbf{z}_n \in \mathbb{R}^D\}_{n=1}^N$, where each slot represents an individual object. Specifically, the mechanism observes the image latent embedding $\mathbf{F}$ to produce an attention readout matrix $\mathbf{U} \in \mathbb{R}^{N \times D_{\text{slot}}}$, where each row $\mathbf{u}_n \in \mathbb{R}^{D_{\text{slot}}}$ of the matrix $\mathbf{U}$ stores values to update $\mathbf{z}_n$ of the slot matrix $\mathbf{Z} \in \mathbb{R}^{N \times D_{\text{slot}}}$. Our framework augments the slot attention mechanism to reflect our additional factorization, which is described in Section 2.1.3.

The decoder employs the Slot Mixer (Sajjadi et al., 2022a), which is a 3D-aware light-field decoder that can identify the spatial extents of objects and account for occlusion. Given a query ray $\mathbf{r} = (\mathbf{o}, \mathbf{d})$, the Slot Mixer generates a feature $\mathbf{x}$ that aggregates features from the object slots. We compute normalized dot-product similarity between $\mathbf{x}$ and $\mathbf{Z}$ using the learned linear projections $W_Q$ and $W_K$, and then use this similarity to compute a weighted mean of the slot matrix $\mathbf{Z}$:

$$\mathbf{w} = \text{softmax}((W_k \mathbf{Z}^\top)^\top (W_q \mathbf{x})), \quad \bar{\mathbf{z}} = \mathbf{w}^\top \mathbf{Z}. \tag{2}$$

From the aggregated slot features $\bar{\mathbf{z}}$, an MLP decoder predicts the RGB value $C(\mathbf{r})$ of the query ray $\mathbf{r}$. The entire network is trained with an $L_2$ reconstruction loss for novel view synthesis, which matches the synthesized images with the ground-truth images $\{\mathbf{I}_i^{\text{gt}}\}$

$$\mathcal{L}_{\text{recon}} = \arg \min_\theta \mathbb{E}_{\mathbf{r} \sim \mathbf{I}_i^{\text{gt}}} \left\| C(\mathbf{r}) - \mathbf{I}_i^{\text{gt}}(\mathbf{r}) \right\|_2^2. \tag{3}$$

### 2.1.2 DECOMPOSITION OF BACKGROUND, FOREGROUND AND AGENTS

Out of the disentangled 3D entities, we clearly identify the active and passive elements to provide the necessary evidence to learn intuitive physics in robotic tasks. Instead of handling the latent vectors $\{\mathbf{z}_n\}$ as a permutation-invariant set, we explicitly assign specific indices to the background slots $\mathbf{z}_{bg}$ and agent slots $\mathbf{z}_{ag}$. The remaining slots are considered as foreground slots $\{\mathbf{z}_{o_n}\}_{n=1}^{N-2}$ which correspond to active objects, and we maintain permutation invariance among object slots. Our training dataset contains the background mask $\mathbf{m}_{bg}^{\text{gt}}$ and agent mask $\mathbf{m}_{ag}^{\text{gt}}$ within image regions from the simulator, which can also be obtained by detecting agents or objects with foundation models. We design an auxiliary loss that matches the background and agent slots to their corresponding masks.

The loss measures the pixel-wise differences between the attention-weighted regions in the predicted image and the regions in the ground truth mask of the RGB input. Formally, the loss is defined as:

$$\mathcal{L}_{\text{bg}} = \sum_{(u,v) \in \Omega} \left\| \mathbf{w}_{bg}(u,v)\,\hat{\mathbf{I}}(u,v) - \mathbf{m}_{bg}^{\text{gt}}(u,v)\,\mathbf{I}(u,v) \right\|_2^2,$$

$$\mathcal{L}_{\text{ag}} = \sum_{(u,v) \in \Omega} \left\| \mathbf{w}_{ag}(u,v)\,\hat{\mathbf{I}}(u,v) - \mathbf{m}_{ag}^{\text{gt}}(u,v)\,\mathbf{I}(u,v) \right\|_2^2. \tag{4}$$

where $\mathbf{I}(u,v) \in \mathbb{R}^3$ is the ground-truth RGB at pixel $(u,v)$, $\hat{\mathbf{I}}(u,v)$ is the predicted RGB, and $\Omega$ is the set of sampled pixel coordinates from the image grid.

Our total loss combines reconstruction and auxiliary mask loss terms

$$\mathcal{L}_{\text{total}} = \mathcal{L}_{\text{recon}} + \lambda_{bg}\mathcal{L}_{\text{bg}} + \lambda_{ag}\mathcal{L}_{\text{ag}}. \tag{5}$$

We can therefore visually observe the physical consequences of various robot interactions and distinguish the actions of robots from other movements. Along with the 3D reasoning, the decomposition can stabilize the training of the robot action policy.

### 2.1.3 3D BLOCK-SLOT ATTENTION

While the slot-based representation interprets the scene into disentangled entities in 3D, the representation is trained only to reconstruct the observed scenes and is limited in generalization. Objects exhibit different properties, such as shape, size, position, or color, which can be categorized and interpreted as a combination of partially shared characteristics. If we can define an extensive set of task goals by combining object attributes, the trained policy can generalize to novel manipulation targets, earning even more sample efficiency.

We propose a novel mechanism called *3D block-slot attention*, which further disentangles the object slots $\mathbf{z}_n \in \mathbb{R}^{D_{\text{slot}}}$ into the concatenation of $M$ blocks of different attributes $\{\mathbf{z}_{n,m} \in \mathbb{R}^{D_{\text{block}}}\}_{m=1}^M$ from a latent vector $\mathbf{F}$ extracted by SRT encoder, such that $D_{\text{slot}} = MD_{\text{block}}$. Since background and agent do not share concepts (e.g., shape, size, position, or color) in the same compositional manner as objects, we adapt the block-slot attention mechanism (Singh et al., 2022) only to the object slots and retain the vanilla slot-attention mechanism for the background and agent slots. By fixing the indices of the agent and background as described in Section 2.1.2, we can design the update mechanism that appropriately mixes block-slot attention and slot attention across all $N$ slots $\mathbf{Z} = \{\mathbf{z}_{bg}, \mathbf{z}_{ag}, \mathbf{z}_{o_1}, \ldots, \mathbf{z}_{o_{N-2}}\}$.

The block-slot attention learns visual concepts by extracting features from images and embedding them into blocks. It extends the slot-attention mechanism with an additional factor binding step, which operates at the block level. To enable this update, each slot update $\mathbf{u}_n$ is equally divided into $M$ segments $\mathbf{u}_{n,m} \in R^{D_{\text{block}}}$. Each block $\mathbf{z}_{n,m} \in R^{D_{\text{block}}}$ is then updated independently using $\mathbf{u}_{n,m}$ with a GRU (Cho et al., 2014), followed by an MLP

$$\mathbf{z}_{n,m} = \text{GRU}_{\phi_m}(\mathbf{z}_{n,m}, \mathbf{u}_{n,m}) \quad \Rightarrow \quad \mathbf{z}_{n,m} \mathrel{+}= \text{MLP}_{\phi_m}(\text{LN}(\mathbf{z}_{n,m})). \tag{6}$$

To make each block retrieve its corresponding representation, we use a concept memory $\mathbf{C}_m \in \mathbb{R}^{K \times d}$ which serves as a learnable soft information bottleneck. The concept memory $\mathbf{C}_m$ consists of $K$ latent prototype vectors associated with the particular factor $m$, which allows each block to perform dot-product attention over the concept memory.

After disentanglement, we can further identify the attributes assigned to specific blocks with additional analysis. For example, if we perform K-means clustering on blocks or swap blocks of different objects, and interpret the semantics of factorized attributes. We observe that the individual blocks represent semantic attributes, such as colors, shapes, sizes, and positions of objects, which are utilized to generalize the policy into novel combinations of scenes.

For non-object slots, $\mathbf{z}_{bg}$ and $\mathbf{z}_{ag}$, we apply vanilla slot-attention updates using their corresponding update vectors $\mathbf{u}_{bg}, \mathbf{u}_{ag} \in R^{D_{\text{slot}}}$, each processed with GRU and MLP independently

$$\mathbf{z}_{n'} = \text{GRU}_{\phi_{n'}}(\mathbf{z}_{n'}, \mathbf{u}_{n'}) \quad \Rightarrow \quad \mathbf{z}_{n'} \mathrel{+}= \text{MLP}_{\phi_{n'}}(\text{LN}(\mathbf{z}_{n'})), \quad \text{where } n' \in \{bg, ag\}. \tag{7}$$

## 2.2 Block Transformer Policy

We propose a block transformer policy that utilizes interpretable attributes in the 3D block-slot representations to efficiently train an agent for a goal-conditioned RL (GCRL) task. We first extract the decomposed slot embeddings using the pre-trained 3D block-slot encoder to represent the current state $s \in \mathcal{S}$ and the goal $g \in \mathcal{G}$ for an RL task. Given the multi-view images for the current state $\{\mathbf{I}_i^s\}$, the pre-trained 3D block-slot encoder extracts the $N$ slots of latent embedding $\mathbf{Z}^s = \{\mathbf{z}_{bg}^s, \mathbf{z}_{ag}^s, \mathbf{z}_{o_1}^s, \ldots, \mathbf{z}_{o_{N-2}}^s\}$, where the object slots $\{\mathbf{z}_{o_i}^s\}$ are further decomposed into $M$ blocks. Similarly, the goal images $\{\mathbf{I}_i^g\}$, provide the latent embedding for the goal state $\mathbf{Z}^g = \{\mathbf{z}_{bg}^g, \mathbf{z}_{ag}^g, \mathbf{z}_{o_1}^g, \ldots, \mathbf{z}_{o_{N-2}}^g\}$. The goal-conditioned policy $\pi^*(s,g)$ maps these structured representations to actions in $\mathcal{A}$ to maximize the reward:

$$\mathbb{E}_\pi[\sum_{t=0}^{\infty} \gamma^t r_t], \quad \text{where } r_t = r(s_t, a_t, g) \tag{8}$$

where $\gamma \in [0, 1)$ is the discount factor, $r : \mathcal{S} \times \mathcal{G} \to \mathbb{R}$ is the reward function designed to minimize the discrepancy between the current and the goal state. By leveraging block-level attributes and agent information in the structured latent embedding, the policy can accurately interpret alignment against the desired goal. Consequently, the policy is efficient in training and achieves stable performance compared to policies employing permutation-invariant OCR (Haramati et al., 2024; Zadaianchuk et al., 2020).

Our block transformer policy employs block-wise cross-attention, instead of object-wise cross-attention, such that the agent focuses on manipulating target objects with correct attributes, as described in Figure 2. We can interpret and combine factorized attributes within the blocks to scrutinize critical relations for the desired policy, which significantly enhances the performance. Suppose the task is to relocate objects with the matched attributes into the goal positions. In that case, we first match objects within the current state and the goal by comparing their semantic attributes using Hungarian matching, while ignoring their positions. Then, we compute block-wise cross-attention between the matched objects. The matched object slot in the current state $\mathbf{z}_{o_n}^s \in \mathbb{R}^{M \times D_{\text{block}}}$ serves as the query, and the corresponding object latent in the goal state $\mathbf{z}_{o_{n'}}^g \in \mathbb{R}^{M \times D_{\text{block}}}$ acts as the keys and values. Then we pool the resulting attention features $\{\mathbf{h}_{n,m}\}_{m=1}^M$ within the object's blocks:

$$\mathbf{H}_n = \text{CrossAttn}(\mathbf{z}_{o_n}^s, \mathbf{z}_{o_{n'}}^g), \quad \mathbf{h}_n = \text{PoolAttn}(\mathbf{H}_n). \tag{9}$$

As the latent structure correctly disentangles components with factorized attributes, the resulting attention feature successfully encodes the pairwise relationship between matched objects in the current and goal states.

In addition to the features extracted from matched objects $\mathbf{h}_n$, the policy incorporates the agent slot features of both the current state $\mathbf{z}_{ag}^s$ and goal state $\mathbf{z}_{ag}^g$, and the current action $\mathbf{a}_t$. Utilizing the agent representation, which is the entity that produces the action, is expected to improve the sample efficiency of vision-based RL (Pore et al., 2024). Additionally, the performance of the policy improves when the attention-based policy network handles the action input separately (Haramati et al., 2024). A self-attention module (Vaswani et al., 2017) aggregates the information to output $\mathbf{P}$, followed by an MLP, which predicts the final output composed of action for the actor and the Q-values for the critic (Figure 1(b)):

$$\mathbf{P} = \text{SelfAttn}([\mathbf{h}_1, \ldots, \mathbf{h}_{N-2}, \mathbf{z}_{ag}^s, \mathbf{z}_{bg}^g, \mathbf{a}_t]), \quad \text{Output} = \text{MLP}(\text{AttnPool}(P)). \tag{10}$$

An extended architectural description of this policy network is provided in Appendix A.1

## 3 Related Work

**3D object-centric learning.** Several recent works have expanded object-centric learning from 2D to 3D representations. They train implicit representations with rendering losses in novel views, inspired by advances in novel view synthesis. OSRT (Sajjadi et al., 2022a) and COLF (Smith et al., 2022) employ an efficient light-field decoder and can quickly decompose complex static scenes. On the other hand, a slot attention module can adapt formulations in neural radiance field, and be trained to reduce rendering losses (Yu et al., 2021; Stelzner et al., 2021; Qi et al., 2023; Luo et al., 2024; Liu et al., 2024). However, all these methods are trained on static scenes without interaction or mutual occlusion, and cannot accurately reason about physical context in dynamic scenes. Our approach additionally provides a highly structured latent representation, enabling systematic reasoning.

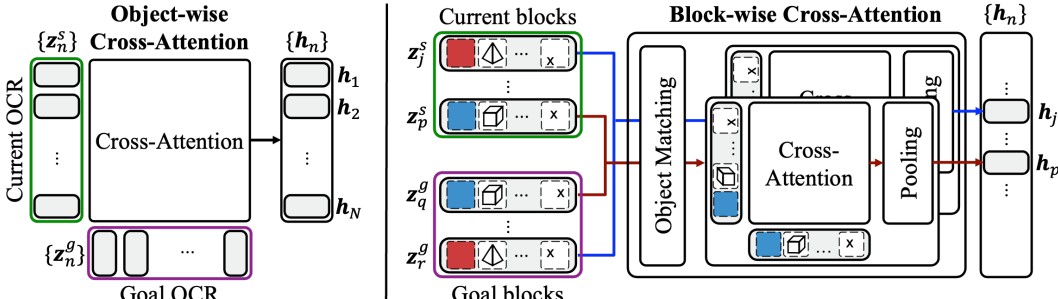

Figure 2: **Comparison between object-wise cross-attention and block-wise cross-attention:** When computing cross-attention between the current and the goal states of the scene, our block transformer policy utilizes a novel structure of decomposed *blocks*, rather than objects. **(a) Object-wise cross-attention** is applied between the set of permutation-invariant slots, without distinguishing objects, agents, and background. **(b) Block-wise cross-attention** operates between blocks, which are consistently decomposed attributes within object slots. By observing the attributes directly, we can stably match corresponding objects in the current and goal representations, and efficiently apply cross-attention to the blocks of desired objects.

**Object-Centric Learning for Structured Representation.** Structured representation methods factor object representations into interpretable attributes (e.g., color, shape, position), which can be composed to produce different combinations of the scenes. SysBinder (Singh et al., 2022) introduces a "block-slot" design to learn such factors in an unsupervised manner. Dreamweaver (Baek et al., 2025) discovers both static and dynamic primitives (e.g., motion direction, speed) for compositional world models in video data. Deep Latent Particles (DLP) (Daniel & Tamar, 2022) and DLPv2 (Daniel & Tamar, 2023) represent scenes as probabilistic particles that encode position and appearance. Including other approaches that factorize object-centric representations (Kosiorek et al., 2018; Stanić & Schmidhuber, 2019; Jiang et al., 2019; Crawford & Pineau, 2019), these approaches often lack explicit 3D reasoning, limiting their robustness to occlusion, viewpoint changes, and real-world interaction.

**Object-Centric Reinforcement Learning.** Object-centric representations in RL enable policies that generalize across diverse scenes with fewer samples. POCR (Shi et al., 2024) integrates "what" (semantic) and "where" (mask) embeddings, improving zero-shot and imitation learning performance in cluttered manipulation tasks. Entity-Centric RL (ECRL) (Haramati et al., 2024) employs DLPv2 perception (Daniel & Tamar, 2023) together with an Entity Interaction Transformer (EIT)–an attention mechanism-based policy architecture– but struggles to recombine attributes with novel objects. PaLM-E (Driess et al., 2023) grounds language models in slot-based scene representations for embodied reasoning, though it falters on unseen property combinations. Additional works (Qi et al., 2024; Watters et al., 2019; Wu et al., 2022; Yoon et al., 2023; Zadaianchuk et al., 2020; 2022; Zhou et al., 2022) enhance RL with structured perception, yet most methods still fail on compositional variations of shape, size, and color, highlighting the need for richer, disentangled, and 3D-aware representations.

## 4 EXPERIMENTS

### 4.1 3D BLOCK-SLOT REPRESENTATION

We first analyze our scene representation, which captures a 3D-aware scene representation with factorized components. For the 3D-awareness, we evaluate the quality of generated novel-view images with PSNR. The object decomposition is evaluated with ForeGround Adjusted Rand Index (FG-ARI) (Hubert & Arabie, 1985; Rand, 1971; Sajjadi et al., 2022a). The additional factorization in block-wise attributes is evaluated in three perspectives: Disentanglement (D), Completeness (C), and Informativeness (I), as proposed in Singh et al. (2022). These values are extracted by training a gradient boost tree algorithm (Locatello et al., 2019) to map the ground-truth factor to the learned representation.

| Dataset | Method | PSNR | FG-ARI | D | C | I |
|---|---|---|---|---|---|---|
| Clevr3D | OSRT | **31.57** | 0.365 | 0.140 | 0.083 | 0.452 |
| | Ours | 31.11 | **0.942** | **0.867** | **0.789** | **0.844** |
| IsaacGym3D | OSRT | **27.35** | 0.321 | 0.403 | 0.222 | 0.769 |
| | Ours | 26.55 | **0.619** | **0.659** | **0.550** | **0.938** |

Table 1: **3D awareness with novel-view synthesis and decomposition performance:** Our method outperforms OSRT across FG-ARI, disentanglement (D), completeness (C), and informativeness (I), while achieving comparable PSNR. The results indicate that our approach improves object decomposition and effectively disentangles information into latent vectors, while maintaining 3D-aware representation.

**Datasets.** We evaluate the performance of scene representation using two datasets: Clevr3D and IsaacGym3D. Clevr3D builds upon the CLEVR dataset (Johnson et al., 2017; Stelzner et al., 2021; Girdhar & Ramanan, 2019), a widely used dataset for object-centric representations. We employ the data generation pipeline (Li et al., 2020; 2021; Lin et al., 2020) to construct Clevr3D, which comprises multi-view observations of multiple objects exhibiting rich combinations of attributes. Each scene consists of a white background and a set of randomly positioned objects, with diverse attributes. Object attributes include three types of shapes (cube, cylinder, sphere), two types of sizes (small, large), and eighteen colors combined with materials (matte, glossy). The cameras are placed randomly on the surface of a hemisphere, capturing six different viewpoints of each scene.

The IsaacGym3D dataset contains more realistic scenarios, in which a robot manipulates objects with randomized actions on a white table. Objects in the scene are randomly assigned one of three types of shapes (cube, cylinder, cone), two types of sizes (small, large), and nine types of colors. They are placed at arbitrary $(x, y)$ positions within the boundaries of the table. The cameras are randomly placed on the surface of a hemisphere to capture diverse perspectives of the environment, resulting in a challenging setting for interpreting agent behavior.

**Results.** We compare the performance of our 3D block-slot representation against OSRT (Sajjadi et al., 2022a), which produces a 3D object-centric representation. They incorporate a similar encoder and decoder architecture, but treat individual components with permutation-invariant slots. In contrast, our approach distinguishes between foreground objects, agents, and backgrounds, and separately handles factorized blocks with block-slot attention. The quantitative results are shown in Table 1. Our results achieve comparable novel-view synthesis results (PSNR) but clearly outperform in distinguishing different components (FG-ARI). The additional factor-binding step enhances the ability to disentangle different attributes in the latent vectors, as indicated by D, C, and I scores, measured using the DCI framework proposed in (Singh et al., 2022). On the other hand, the 3D slot produced by OSRT misses such a structure and fails to assign consistent information to a dedicated location.

In Appendix B.1, we further visualize the information embedded in each block by clustering it based on the values contained within the block. After the interpretable properties, such as color, size, shape, and position, are clearly separated into different blocks, we can utilize the structured representation to manipulate attributes and create a novel combination of the 3D scene. Figure 3 demonstrates exemplary results of novel-view synthesis after swapping blocks between two objects. With the 3D information, we can even emulate realistic occlusion effects. As demonstrated in Section 4.2, such generative capability highly enhances the performance of solving RL tasks. We further provide additional qualitative visualizations, including novel view synthesis, object decomposition, and disentanglement analysis in Appendix B.1. In addition, Appendix B.4 presents ablation studies on the number of blocks, number of prototypes, the auxiliary mask loss, and mixture structure of the slot-attention module and block-slot attention module. Appendix B.2 further provides the single-view inference test results as well as an analysis of the training tendencies.

## 4.2 GOAL-CONDITIONED REINFORCEMENT LEARNING

We exploit the structured representation with the block-transformer policy to perform goal-conditioned RL. The agent must push the objects to the desired positions based on the given goal

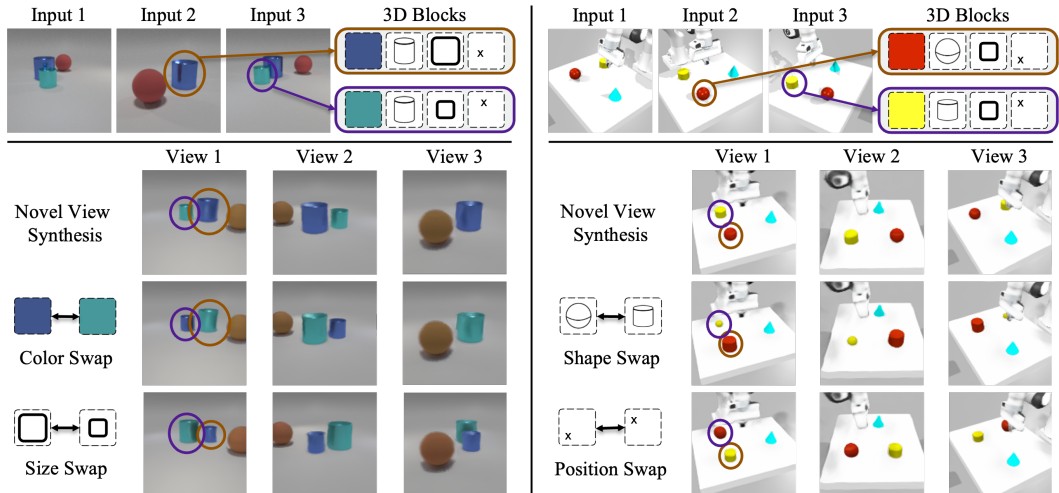

Figure 3: **Novel view synthesis after block manipulation:** After obtaining the structured latent representation of the scene (top), we can manipulate block-wise attributes to create a novel combination of properties in the 3D representations, which can be consistently synthesized in novel view images (bottom). The left images show the results after swapping the color and size blocks in a Clevr3D scene, while the right images are synthesized after swapping the shape and 3D position blocks in the IsaacGym3D dataset.

images. The ground-truth reward is the mean of the negative $L_2$ distance between each object's achieved goal position and its desired goal position. We define *success* as a condition when all $N$ objects are within a threshold $R$ distance from their respective desired goal. The average return is the immediate ground-truth rewards over $T$ timesteps.

We set the RL environment for tabletop object manipulation using the IsaacGym simulator (Makoviychuk et al., 2021). We use the same environmental configuration of IsaacGym3D dataset in Section 4.1, where cameras are fixed to capture the front, left, and right views of a Franka Panda robot arm pushing objects on a white table (Haramati et al., 2024) (see Figure 1, right). In each episode, two objects are randomly respawned on the table with random shapes (cube, cone, cylinder) and colors. We adopt an off-policy RL algorithm, TD3 (Fujimoto et al., 2018) with hindsight experience replay (Andrychowicz et al., 2017). The policy architecture is usable with other on-policy or off-policy algorithms, providing flexibility for various state properties. Appendix A.3 provides additional information on the RL environment configuration.

**Baselines.** We evaluate the performance of three structured scene representations with two policy architectures, including ours. Before training the policy, we first pre-train the scene representation model using the IsaacGym3D dataset. In addition to 3D latent vectors of OSRT in Section 4.1, we also compare against DLPv2 (Daniel & Tamar, 2023), one of the best-performing 2D object-centric representations in goal-conditioned RL. It provides a set of disentangled latent vectors in the form of $P$ foreground particles extracted from a single image $I \in \mathbb{R}^{H \times W \times 3}$. The 2D representations are trained with a separate dataset composed of single-view images. OSRT extracts a set of 3D object-centric latent vector set, $\{\mathbf{z}_n\}$, from multi-view images of a scene $\{\mathbf{I}_i \in \mathbb{R}^{H \times W \times 3}\}$. Implementation details of the pre-training procedures for the baselines are provided in Appendix A.2

The pre-trained visual encoder extracts structured latent embeddings of the current and goal states, which serve as the input to the policy architecture. In addition to the proposed Block Transformer (BT) policy, we adopt the Entity Interaction Transformer (EIT) (Haramati et al., 2024) as a baseline, which is a state-of-the-art object-centric policy for goal-conditioned RL.

**Results.** Our agent observes environments in three categories: in-distribution evaluation (ID), compositional generalization (CG), and out-of-distribution generalization (OOD). The ID refers to the scenes with objects that were observed during training. In contrast, CG and OOD are designed to

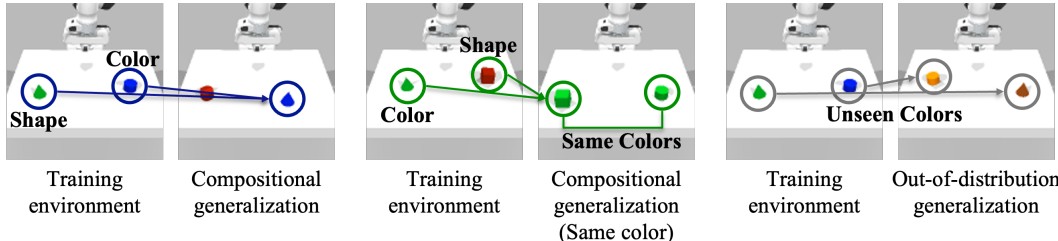

Figure 4: **Evaluation scenarios for compositional and out-of-distribution generalization:** Composition generalization environments consist of objects with properties during training, but novel in their combinations. Out of such unseen combinations, we separately evaluate cases with objects of the same color when the factorization of attributes is unsuccessful. Out-of-distribution environments use objects with colors that were not present in the training set.

| Representation w/ Policy | | DLPv2 w/ EIT | OSRT w/ EIT | Ours w/ EIT | Ours w/ BT |
|---|---|---|---|---|---|
| Success Rate | ID | **0.984** $\pm$ **0.005** | 0.980 $\pm$ 0.011 | **0.984** $\pm$ **0.005** | 0.967 $\pm$ 0.017 |
| | CG | 0.747 $\pm$ 0.030 | 0.758 $\pm$ 0.021 | 0.773 $\pm$ 0.048 | **0.895** $\pm$ **0.011** |
| | CG (same color) | 0.388 $\pm$ 0.064 | 0.414 $\pm$ 0.167 | 0.682 $\pm$ 0.118 | **0.837** $\pm$ **0.035** |
| | OOD | 0.422 $\pm$ 0.170 | 0.700 $\pm$ 0.160 | 0.582 $\pm$ 0.270 | **0.828** $\pm$ **0.099** |
| Avg Return | ID | -0.189 $\pm$ 0.001 | **-0.175** $\pm$ **0.013** | -0.184 $\pm$ 0.014 | -0.218 $\pm$ 0.003 |
| | CG | -0.365 $\pm$ 0.027 | -0.372 $\pm$ 0.002 | -0.382 $\pm$ 0.041 | **-0.310** $\pm$ **0.016** |
| | CG (same color) | -0.594 $\pm$ 0.018 | -0.635 $\pm$ 0.068 | -0.494 $\pm$ 0.096 | **-0.388** $\pm$ **0.016** |
| | OOD | -0.587 $\pm$ 0.090 | -0.414 $\pm$ 0.139 | -0.591 $\pm$ 0.306 | **-0.338** $\pm$ **0.064** |

Table 2: **Performance of goal-conditioned RL:** Our proposed 3D block-slot representation, combined with a block transformer (BT) policy, can effectively interpret goal conditions and exhibit superior performance in various scenarios. We compare the performance of goal-conditioned RL tasks employing different object-centric representations and policy architectures in four settings: ID (in-distribution evaluation), CG (compositional generalization), CG (same color) (compositional generalization with same colored objects), and OOD (out-of-distribution generalization). Results are calculated on 400 randomly sampled goals per seed, with all reported metrics averaged over three random seeds.

analyze the generalization performance of the representation in unseen environments, as depicted in Figure 4. CG uses novel combinations of shapes and colors - the individual factors are encountered during policy training, but not in the same combinations within the same object. OOD includes instances of attributes that were not encountered during policy training. From the OOD cases, we can check how effectively the agent can adapt to zero-shot environments.

Table 2 contains an evaluation of performing goal-conditioned RL tasks with various scene representations and policy architectures. In our model, the agent generalizes well to novel factor compositions, achieving performance comparable to that in the in-distribution evaluation. In contrast, baseline models experience a significant performance drop compared to ID.

Among many failure cases in CG scenarios, we have noticed that sharing the same factor between two objects can severely confuse the instances in the scene. CG (same color) tackles these particularly challenging cases, where the agent must manipulate two objects of the same color. Notably, 3D block-slot representations are highly effective compared to unstructured or 2D representations, as they capture rich visual features within blocks and distinguish same-colored objects, even though such configurations were not seen during training. BT further enhances performance by carefully matching the desired instances with the correct attributes, a task that cannot be achieved with entity-wise attention in EIT. The combination of 3D block-slot representations with the block transformer achieves results comparable to those of ID.

For this evaluation of OOD scenarios, we randomly spawned objects with six unseen colors, all of which were absent in the training distribution. The results show that our model generalizes better

| Generalization Settings | | ID | CG | CG (same color) | OOD |
|---|---|---|---|---|---|
| DLPv2 w/ EIT | ID Multi-View | $0.984 \pm 0.005$ | $0.747 \pm 0.030$ | $0.388 \pm 0.064$ | $0.422 \pm 0.170$ |
| | OOD Multi-View | $0.059 \pm 0.014$ | $0.056 \pm 0.026$ | $0.046 \pm 0.023$ | $0.078 \pm 0.037$ |
| Ours w/ BT | ID Multi-View | $0.967 \pm 0.017$ | $0.895 \pm 0.011$ | $0.837 \pm 0.035$ | $0.828 \pm 0.099$ |
| | OOD Multi-View | $0.948 \pm 0.005$ | $0.877 \pm 0.024$ | $0.818 \pm 0.025$ | $0.865 \pm 0.047$ |
| | ID Single-View | $0.891 \pm 0.004$ | $0.705 \pm 0.023$ | $0.700 \pm 0.038$ | $0.727 \pm 0.105$ |
| | OOD Single-View | $0.802 \pm 0.028$ | $0.726 \pm 0.039$ | $0.676 \pm 0.010$ | $0.758 \pm 0.021$ |

Table 3: **Success rate of view-generalization:** Our model, which leverages a pre-trained 3D block-slot representation and a block transformer (BT), effectively captures 3D object information in a viewpoint-agnostic manner and achieves state-of-the-art performance across diverse generalization settings. We evaluate generalization in goal-conditioned RL tasks under four viewpoints settings: ID Multi-View (in-distribution multi-view), ID Single-View (in-distribution single-view), OOD Multi-View (out-of-distribution multi-view), and OOD Single-View (out-of-distribution single-view). Results are computed over 400 randomly sampled goals per seed, with all reported metrics averaged over three random seeds.

than the other baseline methods in the OOD setting. We report additional RL evaluation results and analysis in Appendix B.3.

Table 3 confirms that our policy can generalize to unseen viewpoints, which implies that it obtains a 3D-aware representation. After training the policy with the front, left, and right camera viewpoints, the view-invariance of the performance is tested on novel viewpoints across multiple generalization scenarios (ID, CG, CG (same color), and OOD). Our method shows only minimal degradation in performance across challenging scenarios, demonstrating strong generalization. In contrast, the 2D baseline with EIT relies on per-view positional encoding, which overfits to view-specific 2D features, and its policy fails under unseen viewpoints.

A notable property of our approach is that it remains effective even when the number and configuration of views at test time differ from training. A policy trained with multi-view observations continues to succeed when provided with only a single training view, and even with a completely novel single-view that appears during training, while still preserving its generalization behavior. This demonstrates that a single-view can provide sufficient information for the policy to infer the necessary 3D-aware object properties. The modest performance drop in the single-view setting is primarily due to robot-induced occlusions, which cannot be fully resolved without multi-view information. Additional results using a suboptimal mask are provided in Appendix B.4.

## 5 CONCLUSION

We propose a 3D block-slot representation, a 3D-aware structured representation that integrates 3D object-centric learning with attribute factorization. We successfully decompose background, foreground, and agent, and can disentangle object factors not only in synthetic static scenes but also in simulated robot environments. The block transformer policy effectively applies structured latent sets to the RL framework, leveraging both the advantages of factor-level input and object-level computation. Compared to prior methods using pre-trained object-centric representations, our policy network improves the performance of compositional generalization and out-of-distribution generalization. The results indicate that the factorized latent structure can be directly used in the policy network, demonstrating superior generalization. While our framework adopts one-to-one static attribute-based matching between current and goal representations, this assumption may be insufficient in scenarios requiring more flexible correspondence strategies, such as matching multiple objects to a single goal. Extending our approach to support more dynamic or many-to-one matching schemes is an important direction for future work, particularly in more complex environments. Furthermore, the 3D block-slot representations, which encode semantics akin to language tokens, have the potential to bridge the gap between 3D perception and Vision-Language-Action (VLA) frameworks.

ACKNOWLEDGEMENTS

This work was supported by Samsung Research Funding & Incubation Center for Future Technology of Samsung Electronics under Project Number SRFC-IT2402-10.

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

APPENDIX

# A    IMPLEMENTATION DETAILS

## A.1    BLOCK TRANSFORMER POLICY.

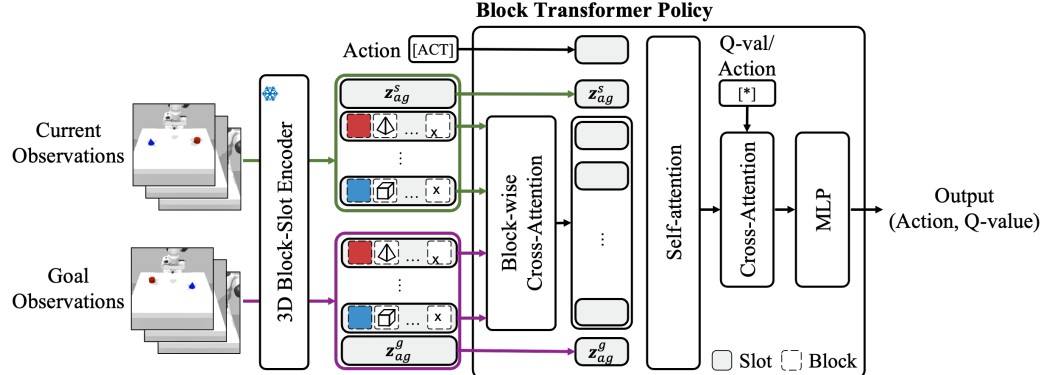

Figure 5:    **Structure of block transformer policy:** The block transformer policy processes 3D block-slot representations using block-wise cross-attention, self-attention, and cross-attention with a pooling token, followed by an MLP to generate the final output.

In this section, we describe the architecture and implementation details of our block transformer policy, including its attention modules and how it integrates information from 3D block-slot representations.

The block transformer policy consists of block-wise cross-attention, self-attention and cross-attention modules, followed by an MLP as described in 2.2. We use only foreground blocks $\{\mathbf{z}_{n,m}\}$ and agent slots $\mathbf{z}_{ag}^s$, $\mathbf{z}_{ag}^g$, as background information tends to be noisy in robotic object manipulation tasks. The self-attention module takes as input the output of block-wise cross attention $\{\mathbf{h}_n\}$, linearly projected action information, and the slots extracted from current and goal observations, in order to model the interaction between agent information and object attributes. The output of self-attention module is then processed through a cross-attention mechanism with a pooling query token, followed by an MLP, to produce the final policy output.

## A.2    PRE-TRAINING REPRESENTATIONS

In this section, we describe the pre-trained representations used in our experiments-DLPv2, OSRT, and our proposed 3D block-slot representations-and introduce the Clevr3D and IsaacGym3D datasets used for training and evaluation. We first describe the datasets collected for pre-training, and then provide details of the 2D and 3D object-centric model.

**Dataset.**    To investigate the capability of our 3D block-slot attention, we use datasets that represent multi-object scenes and robotic environments. First, we consider the CLEVR dataset, which has been widely used to evaluate decomposition & factorization quality. Clevr3D extends CLEVR (Johnson et al., 2017) by randomizing camera positions instead of using fixed viewpoints. Each scene contains three randomly placed objects with attributes independently sampled for shape, size, color and position. Object attributes include three types of shapes (cube, cylinder, sphere), two sizes (small, large), and eighteen colors resulting from combinations of nine base colors with two materials (glossy, matte). The dataset includes 20,000 training scenes (each with six views) and 1,000 test scenes.

In contrast, IsaacGym3D restricts the camera to the front side of the hemisphere, with viewpoints ranging from $-90°$ to $90°$, because the rear views are not necessary for training in the robot manipulation environments. Each scene includes three objects with randomized attributes: shape (cube,

cone, cylinder), size (small, large), and color (identical to Clevr3D). The dataset contains 30,000 training scenes and 200 test scenes. For 2D object-centric learning, we additionally collect two fixed views-front and right views-for each scene.

**Baseline: OSRT.** We train OSRT (Sajjadi et al., 2022a) as the 3D object-centric learning baseline in our experiments. Given multi-view observations $\{\mathbf{I}_i \in \mathbb{R}^{H \times W \times 3}\}$, the SRT encoder aggregates features into a scene-level latent representation $\mathbf{F} \in \mathbb{R}^{L \times D}$. Using the $\mathbf{F}$, the vanilla slot attention module is employed to produce a set of slots $\{\mathbf{z}_n \in \mathbb{R}^D\}_{n=1}^N$. The Slot Mixer decoder then predicts RGB values for a given query ray, as described in Section 2.1. The model is trained with 4 slots of 64 dimensions for Clevr3D, and 6 slots of 64 dimensions for IsaacGym3D. We set hyperparameters listed in Table 6. The model size is intentionally kept small (64-dimensional slots), as larger models-such as those with 1536-dimensional slots-incur significant computational cost and slow inference speed, which is not suitable for vision-based RL.

| Module | Hyperparameter | Clevr3D | | IsaacGym3D | |
|---|---|---|---|---|---|
| | | OSRT | Ours | OSRT | Ours |
| General | Batch size | 40 | 40 | 40 | 40 |
| | Training steps | 1.5M | 1.5M | 1.5M | 1.5M |
| | Learning rate | 0.0001 | 0.0001 | 0.0001 | 0.0001 |
| Slot attention Block-slot attention | Number of slots | 4 | 4 | 4 | 4 |
| | Slot size | 64 | 64 | 64 | 64 |
| | Number of blocks | N/A | 8 | N/A | 8 |
| | Block size | N/A | 8 | N/A | 8 |
| | Number of prototypes | N/A | 20 | N/A | 16 |
| | MLP hidden dimension | 512 | 512 | 512 | 512 |

Table 4: **Hyperparameters of OSRT and 3D block-slot attention used in our experiments.**

**Baseline: DLPv2.** We train DLPv2 (Daniel & Tamar, 2023) as the 2D object-centric learning baseline in our experiments. We follow the modification of DLPv2 by ECRL (Haramati et al., 2024), which sets the background particle features to have a dimensionality of 1, and discards them during RL training. DLPv2 extracts $P$ particles $\mathbf{z}$ from a single image $I \in \mathbb{R}^{H \times W \times 3}$, where $\mathbf{z} = (z_p, z_s, z_d, z_t, z_f) \in \mathbb{R}^{6+m}$. Each latent component-$z_p \in \mathbb{R}^2$, $z_s \in \mathbb{R}^2$, $z_d \in \mathbb{R}$, $z_t \in \mathbb{R}$ and $z_f \in \mathbb{R}^m$-represents the particle's position, scale, depth, transparency and latent feature of visual appearance, respectively. We use the hyperparameters listed in Table 5.

**Baseline: SNeRL.** SNeRL (Shim et al., 2023) is designed to handle dynamic scenes by aggregating multi-view information, including RGB, semantic labels, and distilled features, into a unified representation. However, while this approach is effective for simpler dynamics, our IsaacGym3D setting introduces non-stationary distributions with random object re-sampling, severe clutter, and continuous robot articulation that saturate SNeRL's encoding capacity. Consequently, SNeRL fails to establish stable radiance fields under such high stochasticity as shown in Figure, underscoring the necessity of our transformer-based architecture to robustly model complex manipulation environments.

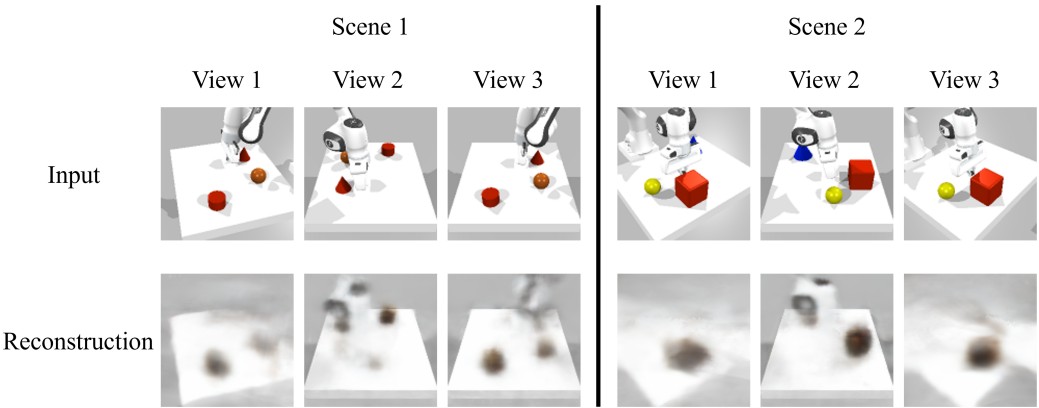

Figure 6: **Qualitative results of SNeRL encoder training on IsaacGym3D:** The failure to reconstruct the input images, particularly the inability to render the robot arm and manipulated objects despite background convergence, demonstrates that SNeRL's encoder cannot effectively capture the complex dynamics and non-stationary distribution of our scene.

### A.3 REINFORCEMENT LEARNING

In this section, we describe the reinforcement learning setup in our experiments, including the reward function, evaluation metrics, and implementation details of reinforcement learning.

| Hyperparameter | DLP |
|---|---|
| Epochs | 60 |
| Batch Size | 64 |
| learning rate | 0.0002 |
| Reconstruction Loss | MSE |
| Prior Patch Size | 16 |
| Posterior KP $P$ | 24 |
| Prior KP Proposals $L$ | 32 |
| Glimpse Size $S$ | 32 |
| Feature Dim $m$ | 4 |
| Background Feature Dim $m_{bg}$ | 1 |
| $\beta_{KL}$ | 0.1 |

Table 5: **Hyperparameters of DLPv2 used in our experiments.**

**Reward.**   The reward is computed from the ground-truth state of the environment and is referred to as the ground-truth reward. It is defined as the mean negative $L_2$ distance between each object and its corresponding goal position on the table:

$$r_{gt,t} = -\frac{1}{N_o} \sum_{i=1}^{N_o} \left\| g_i^d - g_i^a \right\|_2 \tag{11}$$

where each of $r_{gt,t}$, $N_o$, $g_i^d$, and $g_i^a$ denotes the immediate ground-truth reward at time $t$, the number of objects, $i$th object's desired goal, and $i$th object's achieved goal, respectively.

**Evaluation metrics.**   We adopt five evaluation metrics to evaluate the performance of reinforcement learning, following Haramati et al. (2024). Success indicates whether all objects have reached within a given distance threshold from their respective goals. Success fraction measures the proportion of individual objects that satisfy the same distance condition. Average return is the mean cumulative reward over the episode. Maximum object distance refers to the furthest distance between any object and its target at the end of episode, and average object distance is the mean of all

such distances across objects. The formal definitions are as follows:

$$\text{Success} : \mathbb{I}\left(\sum_{i=1}^{N_o} \mathbb{I}(\left\|g_i^d - g_i^a\right\|_2 < R) = N_o\right), \tag{12}$$

$$\text{Success fraction} : \frac{1}{N_o}\sum_{i=1}^{N_o} \mathbb{I}(\left\|g_i^d - g_i^a\right\|_2 < R), \tag{13}$$

$$\text{Average return} : \frac{1}{T}\sum_{t=1}^{T} r_t, \tag{14}$$

$$\text{Maximum object distance} : \max_{i}\{\left\|g_i^d - g_i^a\right\|_2\}, \tag{15}$$

$$\text{Average object distance} : \frac{1}{N_o}\sum_{i=1}^{N_o} \left\|g_i^d - g_i^a\right\|_2. \tag{16}$$

Here, $T$ denotes the number of timesteps in an episode, $r_t$ is the reward at timestep $t$, and $R$ is the distance threshold used to determine success.

**Implementation of RL.** Our reinforcement learning implementation is based on the ECRL (Haramati et al., 2024), built upon Stable-Baseline3 (Raffin et al., 2021), using TD3 (Fujimoto et al., 2018) and HER (Andrychowicz et al., 2017). For exploration, we employ both $\varepsilon$-greedy strategies and Gaussian action noise, following the approach of Zhou et al. (2022). All models are optimized using the Adam optimizer.

| Hyperparameter | EIT & BT |
|---|---|
| Batch size | 512 |
| Learning rate | 5e-4 |
| $\gamma$ | 0.98 |
| $\tau$ | 0.05 |
| Exploration action noise | 0.2 |
| Exploration $\varepsilon$ | 0.3 |
| HER ratio | 0.3 |
| Number of episodes collected per training loop | 16 |

Table 6: **Hyperparameters of RL used in our experiments.**

## B    ADDITIONAL RESULTS

### B.1    QUALITATIVE ANALYSIS OF 3D BLOCK-SLOT REPRESENTATIONS

In this section, we present additional qualitative results to evaluate the effectiveness of our 3D block-slot representation. We include visual comparisons of novel view synthesis and object decomposition between our method and OSRT on the Clevr3D and IsaacGym3D datasets. We also show additional examples of block manipulation through block swapping to demonstrate controllability of the blocks. Furthermore, we analyze factor disentanglement using feature importance matrices and clustering visualizations, which highlight how different blocks encode semantically meaningful components of the scene.

**Novel view synthesis and object decomposition.** We present qualitative comparisons of novel view synthesis results produced by OSRT and our method on the Clevr3D and IsaacGym3D datasets. By leveraging multi-view observations, our method remains robust under occlusions and accurately reconstructs the scene. In particular, our per-slot reconstructions demonstrate more consistent decomposition of background and individual objects across views. Additionally, the robot agent is reliably assigned to a fixed slot index, effectively capturing the agent's presence and action-related information.

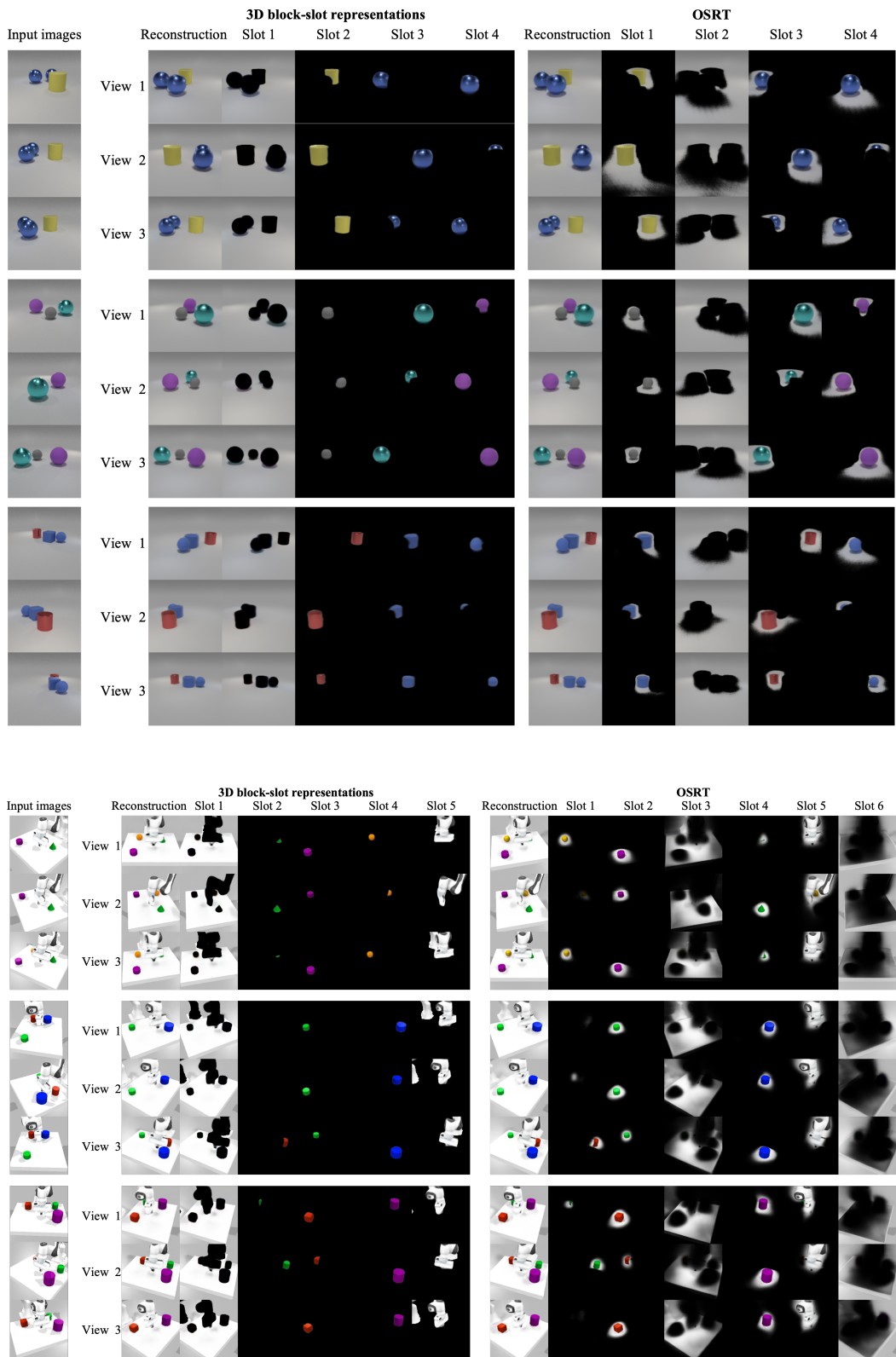

Figure 7: **Novel view synthesis and decomposition:** Our method achieves comparable novel view synthesis quality to OSRT while providing significantly better separation of foreground objects, background and agent components on the Clevr3D and IsaacGym3D datasets.

**Block manipulation and novel view synthesis.** We provide additional visualizations of novel view synthesis through block manipulation, as shown in Figure 3. In each scene, two objects are selected, and the specific blocks corresponding to their attributes are swapped. The model then synthesizes novel views based on the modified slot representations. In both Clevr3D and Isaac-Gym3D, our method successfully generates novel views with swapped blocks, while preserving the consistency of the remaining blocks.

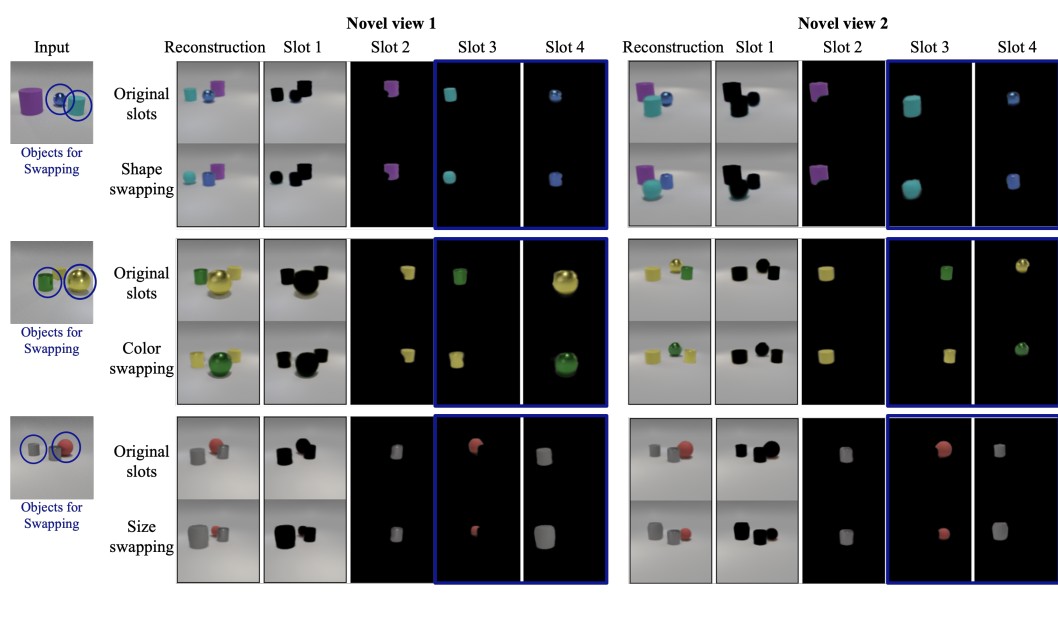

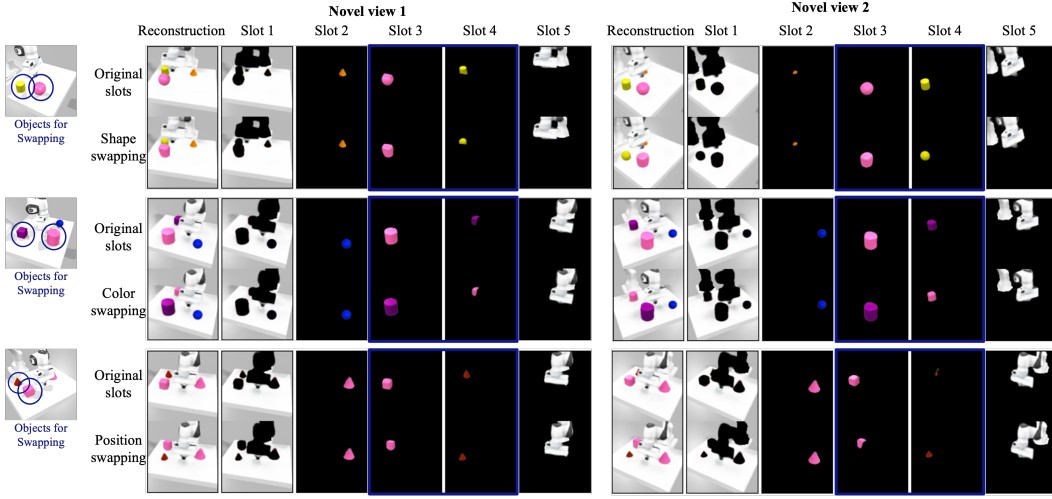

Figure 8: **Additional samples of block manipulation and novel view synthesis:** We show additional examples of block swapping, where blue-circled objects indicate the slots selected for attribute exchange.

**K-means clustering of blocks.** By extending k-means clustering from 2D to 3D block-slot representations, we identify meaningful clusters of blocks. Given multi-view observations $\{\mathbf{I_i}\}$, the 3D block-slot attention mechanism produces 3D block representations $\{\mathbf{z}_{n,m}\}$. Unlike the 2D slot attention mechanism, our model aggregates multi-view information into latent vectors and thus cannot directly produce attention maps from the encoder outputs. Therefore, we obtain the corresponding

object images $\{\mathbf{y}_n\}$ using the slot weights $\mathbf{w}$ from Slot Mixer decoder applied to a novel view. Using 1,050 objects in 350 scenes, each containing three independently randomized objects, we collect a set of pairs $\{(\mathbf{z}_{n,m}, \mathbf{y}_n)\}$. K-means clustering of the blocks (with $K = 10$) reveals the underlying structure of the representation helps identify which blocks encode specific object attributes. In block 6 of the Clevr3D dataset, cluster 1 corresponds to cylindrical objects, while cluster 3 corresponds to spherical objects. These results suggest that block 6 encodes shape-related attributes.

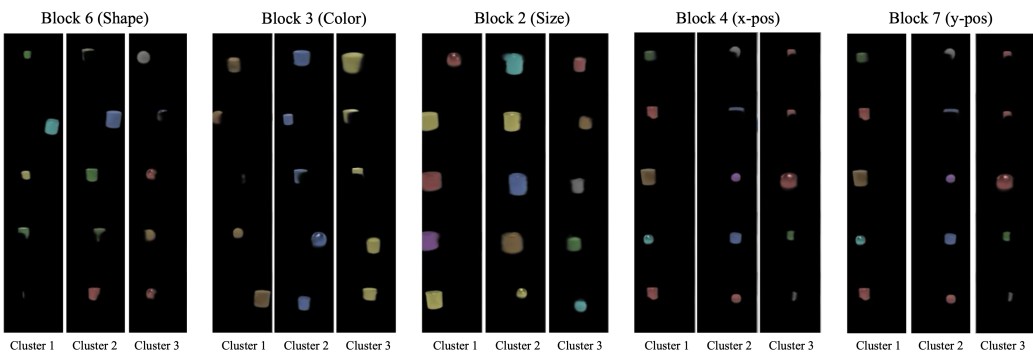

Figure 9: **K-means clustering of 3D block representations:** Clustering results indicate that different blocks capture distinct object attributes. For example, block 6 clustered by shape, block 3 by color, block 2 by size, block 4 by x-position, and block 5 by y-position.

**Feature importance matrix.** We visualize the feature importance matrices for our method and OSRT. As described in Section 4.1, we train a gradient boost tree to predict each ground-truth object attribute from the learned blocks, in order to evaluate how well the blocks capture and disentangle object attributes. Using the trained model, we extract the feature importance matrix, where each row corresponds to a ground-truth object factor, and each column represents a latent unit-either a slot in OSRT or a block in our method. The results show that our model learns more disentangled representations, with different blocks specializing in distinct object attributes across both the Clevr3D and IsaacGym3D datasets. In contrast, the slot-based representations from OSRT show more entangled feature importance patterns, indicating less structured encoding of object attributes.

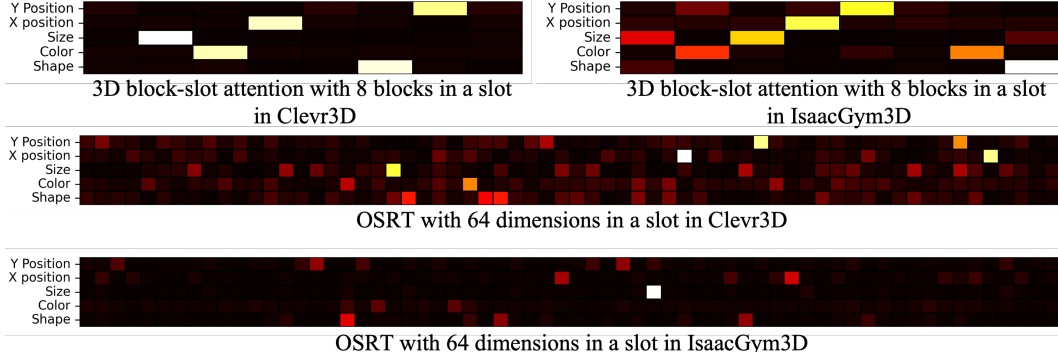

Figure 10: **Feature importance matrix:** Feature importances for predicting ground-truth object attributes from latent representations. Our method shows clear attribute-specific encoding in each block, while OSRT's 3D slot representations are less disentangled.

## B.2 QUANTITATIVE ANALYSIS OF 3D BLOCK-SLOT REPRESENTATIONS

In this section, we provide additional quantitative results to further demonstrate the 3D-awareness of our 3D block-slot representation. Specifically, we evaluate how a model trained under the multi-view setting on the IsaacGym3D dataset performs when only a single view is provided at inference time.

**Single-view inference performance.** Table 7 reports the performance of our 3D block-slot representation, which is trained with three input views in the multi-view setting but evaluated with only a single input view. The results show that the performance with a single view is nearly identical to that obtained with three views. The slight decrease in PSNR and DCI is expected, as occlusions are more challenging to resolve in the single-view setting compared to the multi-view setup.

| View settings | PNSR | FG-ARI | D | C | I |
|---|---|---|---|---|---|
| Multi-View | 26.55 | 0.619 | 0.659 | 0.550 | 0.938 |
| Single-View | 25.90 | 0.623 | 0.626 | 0.558 | 0.915 |

Table 7: **Novel-view synthesis and decomposition performance under multi-view and single-view settings:** Our model is trained in the multi-view setting, and the table compares its inference performance when using multiple views versus a single view.

**Training loss behavior.** As shown in Figure 11, our model's training objective consists of a reconstruction loss and an auxiliary mask loss. The reconstruction loss exhibits two notable drops during training. These correspond to the model first learning to synthesize the agent correctly from novel views using the agent slot, and later learning to reconstruct the objects through the object slots. After these two transitions, both the reconstruction loss and the mask losses decrease smoothly and stabilize over time. Importantly, block-wise separation also emerges shortly after successful object reconstruction, and the blocks become increasingly disentangled as training progresses.

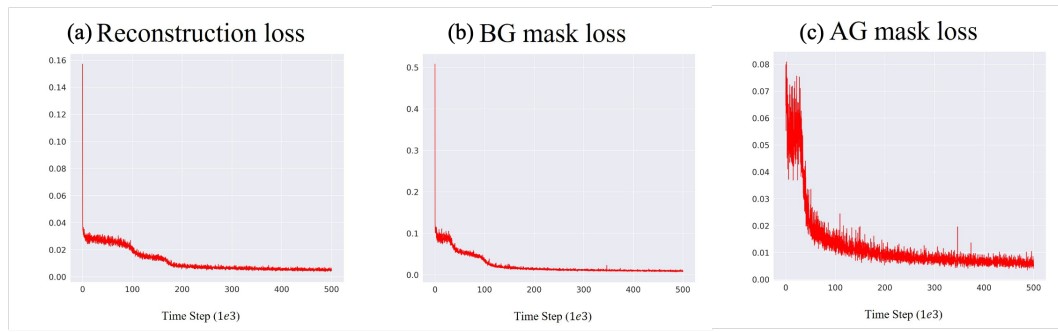

Figure 11: **Train loss of 3D block-slot representation:** We show examples of suboptimal masks: background masks derived from DINO attention maps in simulation, and agent masks obtained from hand–eye calibration and robot kinematics in real-world settings.

## B.3 ADDITIONAL RL RESULTS

In this section, we present additional reinforcement learning results to complement the main experiments. We include agent rollout visualization, additional evaluation metrics, and learning curves. These results provide a more comprehensive understanding of our policy's performance and generalization capabilities. In addition, we report results from non-object-centric baselines to clearly contrast them with object-centric representations and highlight the necessity of pre-trained object-centric representations.

**Rollouts of an agent across evaluation scenarios.** We assess the generalization capability of our model across four different environment setups: in-distribution evaluation, compositional generalization, compositional generalization of same-color objects, and out-of-distribution generalization.

As reported in Table 2, our method demonstrates strong performance across all scenarios. The visualized rollouts show that the agent successfully completes the task in all evaluation scenarios, further demonstrating the robustness and generalization ability of our policy.

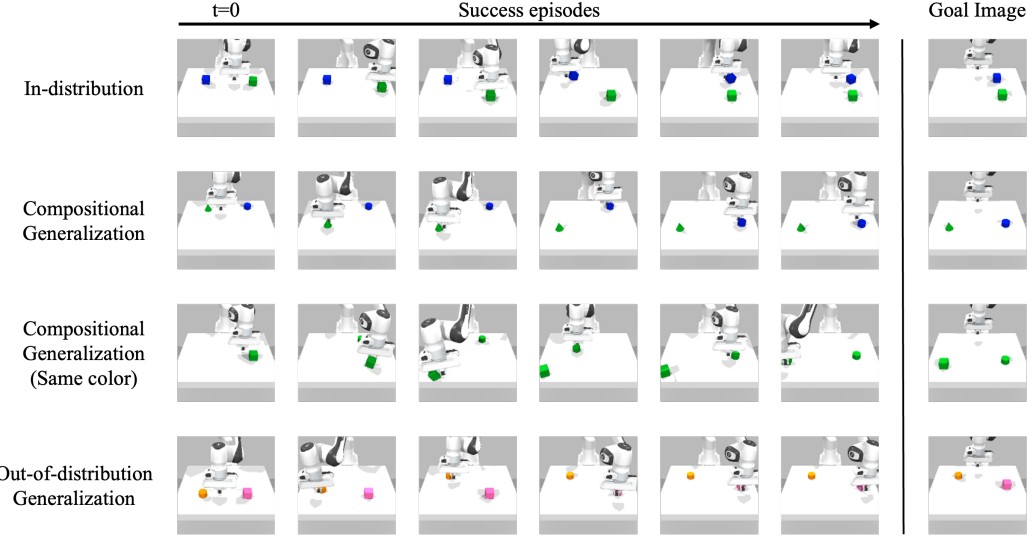

Figure 12: **Agent behavior in evaluation tasks:** Visualized rollouts demonstrate that the agent completes the task successfully across all evaluation scenarios, including in-distribtution, compositional, and out-of-distribution settings.

**Additional evaluation metrics.** We evaluate the performance of four methods that combine pretrained representations with reinforcement learning policies. Following (Haramati et al., 2024), we adopt five evaluation metrics: success rate, success fraction, average return, maximum object distance, and average object distance. The results of success rate and average return are reported in Table 2, while the remaining metrics are shown in Table 8. Success fraction measures the proportion of individual objects that reach their respective goal positions. Maximum and average object distances are computed based on the Euclidean distances between each object and its assigned goal. As shown in Table 8, our method consistently outperforms all baselines across the additional metrics as well. Specifically, it achieves higher success fraction and lower object distance errors, indicating more precise and consistent goal-reaching behavior. These results further support the effectiveness of our policy and its strong generalization capability across different evaluation settings. Formal definitions of these metrics are provided in Appendix A.3

| Representation w/ Policy | | DLPv2 w/ EIT | OSRT w/ EIT | Ours w/ EIT | Ours w/ BT |
|---|---|---|---|---|---|
| | ID | **0.991** ± 0.003 | 0.988 ± 0.008 | 0.990 ± 0.003 | 0.979 ± 0.013 |
| Success Fraction | CG | 0.843 ± 0.016 | 0.843 ± 0.003 | 0.833 ± 0.037 | **0.913** ± 0.010 |
| | CG (same color) | 0.628 ± 0.038 | 0.598 ± 0.073 | 0.750 ± 0.091 | **0.866** ± 0.027 |
| | OOD | 0.654 ± 0.113 | 0.812 ± 0.107 | 0.720 ± 0.192 | **0.888** ± 0.061 |
| | ID | 0.026 ± 0.001 | **0.021** ± 0.003 | 0.022 ± 0.001 | 0.030 ± 0.003 |
| Max Obj Dist | CG | 0.084 ± 0.010 | 0.083 ± 0.005 | 0.083 ± 0.016 | **0.049** ± 0.005 |
| | CG (same color) | 0.167 ± 0.012 | 0.172 ± 0.037 | 0.108 ± 0.034 | **0.066** ± 0.004 |
| | OOD | 0.164 ± 0.034 | 0.102 ± 0.053 | 0.164 ± 0.113 | **0.059** ± 0.019 |
| | ID | 0.018 ± 0.001 | **0.016** ± 0.002 | 0.017 ± 0.002 | 0.023 ± 0.002 |
| Avg Obj Dist | CG | 0.054 ± 0.005 | 0.054 ± 0.002 | 0.056 ± 0.009 | **0.039** ± 0.003 |
| | CG (same color) | 0.103 ± 0.006 | 0.114 ± 0.017 | 0.081 ± 0.021 | **0.052** ± 0.003 |
| | OOD | 0.100 ± 0.021 | 0.064 ± 0.031 | 0.105 ± 0.072 | **0.042** ± 0.013 |

Table 8: **Additional evaluation metrics for the IsaacGym push task:** We report results across four settings: ID (in-distribution evaluation), CG (composition generalization), CG-same (compositional generalization with same colored objects ), and OOD (out-of-distribution generalization). Each metric is averaged over 400 randomly sampled goals per seed.

**Visualization of GCRL training.** We employed four methods that combine pre-trained vision models with control policies in our setup. We visualize both the success rate and the average reward over training timesteps. Among these methods, the combination of 3D block-slot representations with an object-centric policy achieves the highest sample efficiency. This suggests that the structured 3D block representations are more effective for learning robotic manipulation tasks than unstructured 3D object-centric representations or 2D object-centric representations. In addition, our block-level transformer, while slightly less sample-efficient, demonstrates superior generalization performance in environments where the object-centric policy fails.

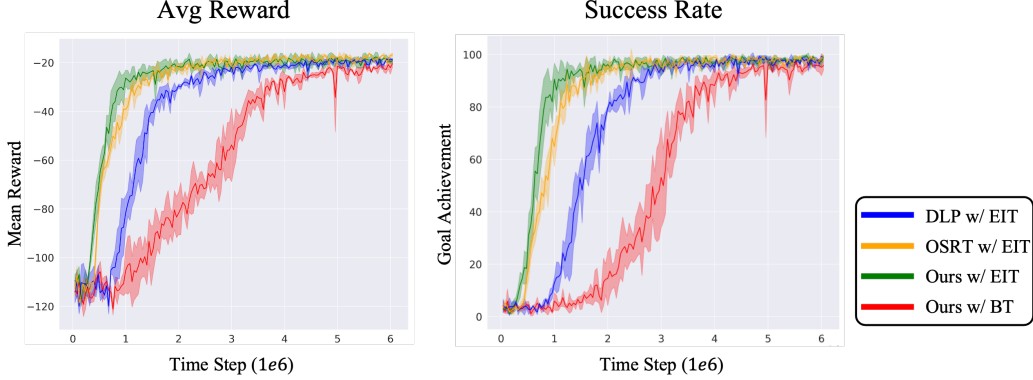

Figure 13: **Success rate and average reward during GCRL training:** Comparison of four methods using different pre-trained representations and policies. The 3D block-slot representation with object-centric policy achieves the fastest learning, and the block-level transformer shows stronger generalization in long-horizon tasks.

**Additional results on non-object-centric baseline.** Following the non-object-centric baseline used in ECRL (Haramati et al., 2024), we evaluate a policy that encodes multi-view state and goal images into a single latent vector using a pre-trained autoencoder (Nair et al., 2018), which is subsequently processed by a standard MLP policy. As reported in Table 9, this baseline does

no reliably solve multi-object manipulation tasks. Compressing the entire scene into unstructured latent representations make it difficult to preserve object identities, spatial relationships, and goal correspondence, resulting in unstable learning behavior. In contrast to object-centric approaches, the unstructured latent bottleneck prevents the agent from explicitly perceiving individual objects and reasoning about their interactions, highlighting a fundamental limitation of unstructured visual representations for multi-object control.

| Model | Success Rate | Avg Return | Success Fraction | Max Obj Dist | Avg Obj Dist |
|---|---|---|---|---|---|
| VAE w/ MLP | $0.042 \pm 0.015$ | $-1.072 \pm 0.040$ | $0.185 \pm 0.004$ | $0.273 \pm 0.010$ | $0.210 \pm 0.008$ |
| Ours w/ BT | $0.967 \pm 0.017$ | $-0.218 \pm 0.003$ | $0.979 \pm 0.013$ | $0.030 \pm 0.003$ | $0.023 \pm 0.002$ |

Table 9: **Performance of goal-conditioned RL of non-object-centric baseline and ours:** We report the performance of a non-object-centric baseline using a VAE-based visual representation with an MLP policy, and our object-centric method combined with a BT policy.

### B.4 ADDITIONAL ABLATION STUDIES

In this section, we present additional ablation experiments. We investigate the effect of the number of blocks and prototypes in the 3D block-slot attention module, as well as the impact of the auxiliary mask loss and its scaling. We also report GCRL results using suboptimal masks, showing that our model maintains strong performance even without ground-truth masks. Finally, we ablate the 3D block-slot attention architecture by comparing vanilla slot attention and block-slot attention, demonstrating the necessity of combining the two mechanisms.

**Mixture structure of slot attention.** We evaluate the contribution of the slot-attention module and the block-slot attention module used in our model for decomposing the background, agent, and objects. In our architecture, the background and agent slots are updated using the vanilla slot-attention module, whereas the object slots are updated using the block-slot attention module, which enables learning object attributes in a block-wise manner. In contrast, Figure 14 and Table 10 presents the results obtained when all slots are updated either with the vanilla slot-attention module or with the block-slot attention module.

When all slots are updated solely with vanilla slot-attention, which corresponds to applying our auxiliary mask loss to an OSRT, the background and agent can be assigned to consistent slots, but overall performance remains clearly inferior. This variant shows lower PSNR, FG-ARI, and DCI scores compared to our mixed architecture and, importantly, fails to disentangle object attributes into block-wise latent components. As a result, it cannot provide the structured 3D block-slot representations required by the block transformer policy that enables strong generalization. These findings indicate that vanilla slot-attention, even with auxiliary mask loss, is insufficient, and that our mixed design is necessary for accurate decomposition and attribute-aware reasoning.

Conversely, when all slots are updated using the block-slot attention module, the background and agent—which do not share the same attribute structure as objects—are forced to learn object-like attributes. This causes the model to fail at both correct object decomposition and block-wise attribute learning. These observations indicate that background and agent slots, which do not share object attributes such as color, shape, size, and position, must follow update pathways. Our combination of the vanilla slot-attention module for background and agent slots, the block-slot attention module for object slots, and the auxiliary mask loss enables successful decomposition of the background, agent, and objects in robotic scenes, while also supporting block-wise attribute learning for each object.

| Model | PSNR | FG-ARI | D | C | I |
|---|---|---|---|---|---|
| Ours | 26.55 | 0.619 | 0.659 | 0.550 | 0.938 |
| Ours with only Slot-Attention | 25.28 | 0.520 | 0.501 | 0.302 | 0.786 |

Table 10: **Comparison of slot-attention and our mixture slot-attention architecture:** We evaluate all slots are updated solely with the vanilla slot-attention module. The results are compared against the proposed mixture slot-attention, which applies vanilla slot-attention to the background and agent slots while updating foreground slots using the block-slot attention module..

| Input image 1 | Input image 2 | Input image 3 | Novel View 1 GT | Novel View 1 Reconstruction | Novel View 2 GT | Novel View 2 Reconstruction |

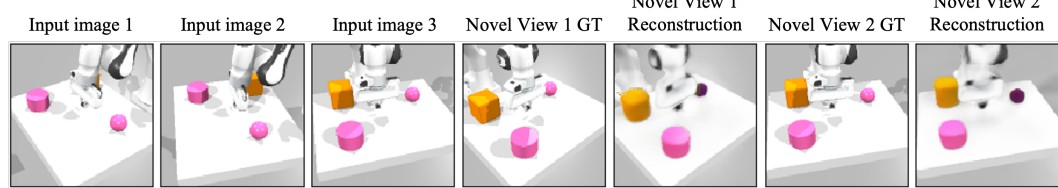

Figure 14: **Failure cases when updating all slots with the block-slot attention module:** When the block-slot attention module is applied to all slots—including background and agent slots—the model fails to learn stable attributes, leading to training collapse and incorrect scene decomposition.

**Number of blocks.** We evaluate the performance of the 3D block-slot representation under different numbers of blocks using three metrics: PSNR, FG-ARI, and DCI performance. Each model is trained with 4, 8, 16 blocks for approximately 600k iterations, while keeping the slot size fixed to 64; consequently, the block sizes are 16, 8, and 4, respectively. The results show a clear trend: increasing the number of blocks leads to noticeably improved disentanglement. In other words, as the number of blocks increases and the block size decreases, the model tends to assign more independent properties to individual blocks. This behavior arises because smaller block sizes impose stronger constraints on the representational capacity of each block, while having more blocks provides the model with greater flexibility to distribute and separate different object attributes.

| Number of blocks | PSNR | FG-ARI | D | C | I |
|---|---|---|---|---|---|
| 4 | 24.45 | 0.498 | 0.322 | 0.400 | 0.844 |
| 8 | 23.71 | 0.465 | 0.429 | 0.394 | 0.836 |
| 16 | 24.19 | 0.437 | 0.447 | 0.368 | 0.848 |

Table 11: **Effect of the number of blocks on 3D awareness performance:** We provide PSNR, FG-ARI, and DCI performance as a function of the number of blocks.

**Number of prototypes.** We evaluate and analyze the performance of the 3D block-slot representation with varying numbers of prototypes using three metrics: PSNR, FG-ARI, and DCI performance. Each model is trained for approximately 1M iterations, differing only in the number of prototypes, which is set to 8, 16, or 32. When using 8 prototypes, the 3D block-slot representation successfully performs novel-view synthesis and decomposes scenes into per-object slots, but it fails to disentangle object attributes across blocks. The number of prototypes directly influences the representational capacity of each block; when the number of prototypes is too small, the model cannot reliably separate the underlying 3D attributes. This trend is consistently observed in the models trained with 16 and 32 prototypes. Models with more prototypes not only achieve higher PSNR but also exhibit substantially improved disentanglement performance. These results indicate that increasing the number of prototypes enhances the expressive power of each block, allowing individual blocks to more fully encode distinct attributes.

| Number of prototypes | PSNR | FG-ARI | D | C | I |
|---|---|---|---|---|---|
| 8 | 22.43 | 0.654 | 0.023 | 0.030 | 0.476 |
| 16 | 24.86 | 0.491 | 0.454 | 0.439 | 0.916 |
| 32 | 25.33 | 0.495 | 0.480 | 0.432 | 0.876 |

Table 12: **Effect of the number of prototypes on 3D awareness performance:** We provide PSNR, FG-ARI, and DCI performance as a function of the number of prototypes.

**Impact of block disentanglement.** We evaluate the GCRL performance of a 3D block-slot representation with relatively low DCI scores. For this comparison, we use a checkpoint with incomplete disentanglement, where the model can already decompose scenes into background, objects, and agent, but the block-level disentanglement is still incomplete, and compare its success rates across

four generalization settings: ID, CG, CG (same color), and OOD. As shown in Table 13, the model with insufficient block separation exhibits consistently lower success rates across all four settings. This degradation arises because the dynamic and static properties are not clearly disentangled, making the Hungarian matching—performed by the block transformer policy based on static attributes— less reliable. Furthermore, the block-wise cross-attention receives weaker and less distinct signals, which makes it harder for the policy to learn task-relevant object relations to generalize effectively.

| Model | ID | CG | CG (same color) | OOD |
|---|---|---|---|---|
| High-DCI Model | $0.967 \pm 0.017$ | $0.895 \pm 0.011$ | $0.837 \pm 0.035$ | $0.828 \pm 0.099$ |
| Low-DCI Model | $0.823 \pm 0.015$ | $0.677 \pm 0.030$ | $0.599 \pm 0.037$ | $0.750 \pm 0.000$ |

Table 13: **Impact of disentanglement quality on GCRL performance:** We compare success rates across four generalization settings (ID, CG, CG (same color), and OOD) between a high-DCI model and a low-DCI model.

**Auxiliary mask loss scale.** We evaluate and analyze the performance of the 3D block-slot representation under different scales of the auxiliary mask loss introduced in Eq 4, using three metrics: PSNR, FG-ARI, and DCI performance. Each model is trained for approximately 700k iterations with the auxiliary mask loss weights $\lambda_{bg}$ and $\lambda_{ag}$ set to 0.1, 0.5, or 1.0. Compared to the models trained with $\lambda_{bg}$ and $\lambda_{ag} = 0.1$ or 0.5, the model using a scale of 1.0 achieves higher FG-ARI but shows lower PSNR, disentanglement, completeness, and informativeness scores. This indicates that when the auxiliary mask loss becomes too dominant relative to the primary reconstruction loss $\mathcal{L}_{recon}$—the key objective responsible for novel-view synthesis and unsupervised disentanglement of object attributes into separate blocks—the model focuses more on accurate decomposition at the expense of view synthesis quality and attribute embedding fidelity. In other words, excessively large $\lambda_{bg}$ and $\lambda_{ag}$ can hinder 3D block disentanglement. Nevertheless, the degradation is modest, and more importantly, the inclusion of the auxiliary mask loss—regardless of the specific scaling— enables the model to reliably distinguish background, foreground, and agent regions, which is essential for downstream block-slot decomposition.

| $\lambda_{bg}, \lambda_{ag}$ | PSNR | FG-ARI | D | C | I |
|---|---|---|---|---|---|
| $\lambda_{bg}, \lambda_{ag} = 0.1$ | 24.66 | 0.426 | 0.472 | 0.446 | 0.856 |
| $\lambda_{bg}, \lambda_{ag} = 0.5$ | 24.39 | 0.425 | 0.450 | 0.434 | 0.858 |
| $\lambda_{bg}, \lambda_{ag} = 1.0$ | 23.62 | 0.457 | 0.407 | 0.382 | 0.828 |

Table 14: **Effect of the scale of auxiliary mask loss on 3D awareness performance:** We provide PSNR, FG-ARI, and DCI performance as a function of the scale of auxiliary mask loss.

**Suboptimal mask.** We evaluate the performance of our method trained with suboptimal masks— rather than ground-truth masks—across four generalization settings (ID, CG, CG (same color), and OOD), as well as under training views, randomized novel views, and single-view inputs. Because our approach requires separating the agent, background, and remaining objects, a natural concern is whether high-quality masks are necessary. However, we emphasize that our method can operate effectively without direct access to GT agent or background masks. For the agent mask, given the robot's URDF and joint readings, we reconstruct its current visual appearance, and render the corresponding agent mask for any desired viewpoint with a rasterizer using the camera pose obtained via standard hand-eye calibration. For the background mask, we acknowledge the inherent ambiguity in defining what constitutes background. To address this, we obtain coarse foreground masks using DINO (Oquab et al., 2023; Caron et al., 2021) by normalizing the principal component of its feature map; their complements provide coarse background masks. Leveraging DINO's large-scale self-supervised pre-training allows us to reliably distinguish foreground from background despite the lack of explicit labels. By combining these components, we obtain suboptimal background and agent masks for training. Representative examples of the real-world agent masks and simulator-derived background masks are shown in Figure 15.

As shown in Table 15, the GCRL performance of our model trained with suboptimal masks is closely approach that of the model trained with ground-truth masks. This holds consistently across all four generalization settings—ID, CG, CG (same color), and OOD—and remains true even when evaluated under OOD randomized views or when provided with only a single input view. In other words, our model is capable of decomposing background, foreground, and agent regions using only suboptimal masks, without requiring any access to ground-truth segmentations, while still achieving GCRL performance comparable to the ground-truth-trained counterpart. This demonstrates that our approach offers strong 3D-aware, view-agnostic, attribute-centric generalization, enabled solely by suboptimal masks that are readily obtainable in both simulation and real-world scenarios.

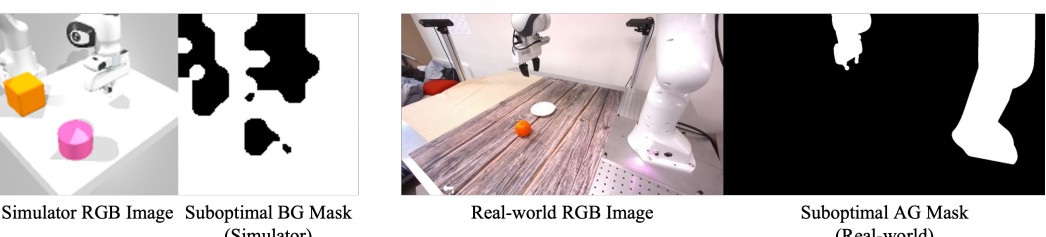

| Simulator RGB Image | Suboptimal BG Mask (Simulator) | Real-world RGB Image | Suboptimal AG Mask (Real-world) |

Figure 15: **Examples of suboptimal mask:** We show examples of suboptimal masks: background masks derived from DINO attention maps in simulation, and agent masks obtained from hand–eye calibration and robot kinematics in real-world settings.

| Generalization Settings | | ID | CG | CG (same color) | OOD |
|---|---|---|---|---|---|
| Ours w/ GT masks | ID Multi-View | $0.967 \pm 0.017$ | $0.895 \pm 0.011$ | $0.837 \pm 0.035$ | $0.828 \pm 0.099$ |
| | OOD Multi-View | $0.948 \pm 0.005$ | $0.877 \pm 0.024$ | $0.818 \pm 0.025$ | $0.865 \pm 0.047$ |
| | ID Single-View | $0.891 \pm 0.004$ | $0.705 \pm 0.023$ | $0.700 \pm 0.038$ | $0.727 \pm 0.105$ |
| | OOD Single-View | $0.802 \pm 0.028$ | $0.726 \pm 0.039$ | $0.676 \pm 0.010$ | $0.758 \pm 0.021$ |
| Ours w/ suboptimal masks | ID Multi-View | $0.974 \pm 0.023$ | $0.771 \pm 0.000$ | $0.880 \pm 0.037$ | $0.891 \pm 0.066$ |
| | OOD Multi-View | $0.943 \pm 0.037$ | $0.870 \pm 0.022$ | $0.813 \pm 0.029$ | $0.870 \pm 0.052$ |
| | ID Single-View | $0.917 \pm 0.000$ | $0.771 \pm 0.088$ | $0.703 \pm 0.007$ | $0.766 \pm 0.022$ |
| | OOD Single-View | $0.854 \pm 0.044$ | $0.730 \pm 0.015$ | $0.682 \pm 0.066$ | $0.730 \pm 0.015$ |

Table 15: **Success rate of goal-conditioned RL for our models with GT masks versus suboptimal masks:** Our 3D block-slot representation supports stable slot assignments even when trained with suboptimal masks instead of GT masks. To evaluate this, we compare the goal-conditioned RL success rates of our model trained with GT masks and with suboptimal masks across four environment generalization settings (ID, CG, CG (same color), OOD) and four view settings (ID Multi-View, ID Single-View, OOD Multi-View, OOD Single-View)

## C  LLM USAGE

We use a large language model (LLM) to detect and correct grammatical errors and awkward phrasing.

