# OpenReview forum: "3D-aware Disentangled Representation for Compositional Reinforcement Learning"
_ICLR.cc/2026/Conference — ICLR 2026 Poster_

### Official Review · Reviewer_xPHR · 2025-10-18

**Soundness:** 3
**Presentation:** 3
**Contribution:** 3
**Rating:** 6
**Confidence:** 4

**Summary:**

This work proposes a 3D-aware object-centric representation of scenes for training Reinforcement Learning (RL) agents that exhibits improved generalization capabilities to unseen compositions of object-level attributes in goal-conditioned robotic manipulation tasks. The representation is based on the 3D object scene representation transformer (OSRT) and incorporates the block-slot attention module for further decomposition of slots into object-level factors. The representation is evaluated on Clevr3D and a novel simulated robotic manipulation dataset and improves over OSRT in various metrics. The paper additionally proposes a policy architecture for a goal-conditioned RL task that leverages the object-level factorization of the representation to achieve better zero-shot transfer to unseen compositions/variations of object-level attributes compared to baselines.

**Strengths:**

**Overview**
- Novelty and contribution are clear.
- Self-contained and clear method description.
- Learned representation facilitates human interpretability and controllability.
- Good empirical results in terms of transfer to unseen test tasks.
- Thorough study of the different method components.
- Detailed and helpful figures.
- Detailed Appendix.

I am positive about raising my score if at least some of the concerns I have raised in the Weaknesses/Questions sections are addressed.

**Weaknesses:**

**Overview**
- Abstract and Intro are not clear and logic is hard to follow.
- Assumes access to agent and background masks.
- The Block Transformer relies on human annotation of blocks within the proposed block-slot representation (post representation training but prior to RL policy training).
- Incorporates multiple assumptions about the specific task in the experiment within the RL Block Transformer architecture, which may limit its wider applicability.

**Abstract**

After reading the paper and understanding it, I still find the abstract very confusing. It uses terms which I would not consider as common knowledge such as “shared concept memory” and some misleading phrases such as “The representation, therefore, can stably identify and track 3D trajectories”---reading this I would assume the representation operates on sequences of images (i.e., videos) but it is a multi-view image representation if I am not mistaken. I suggest making the abstract more “abstract” to appeal to the general reader.

**Introduction**

The two key challenges presented in the introduction for “fully utilizing object-centric representations” (lines 52-53) should be backed-up by citations or by results in this work but these are not referenced. This paragraph reads more like—“these two aspects of our method are what is missing”---but the challenges and their causes are not clearly described.

**Assumptions and Limitations**

*Agent and background masks*: this assumption makes the overall method no longer unsupervised. That in itself is not a problem but could introduce a limitation which stems from reliance on humans to e.g., define what is considered the “background” as opposed to being learned from data. It seems that the method strongly relies on the assumption that these masks can be obtained with high fidelity which may make the overall pipeline brittle when this is not the case. The authors claim that these masks can be obtained using foundation models. If so, why not obtain the entire object-centric representation using foundation models? Is your representation-learning method applicable without the dedicated background and agent slots?

*Identifying the meaning of latent block attributes*: your method seems to produce human-interpretable disentanglement in the latent slot representations of scenes considered in this work, which is impressive and a positive outcome in my opinion. That said, your RL policy architecture relies on the human interpretation of the role of each block, which is another form of supervision that can make scaling difficult and introduce brittleness.

*Strong assumptions about a specific task embedded within the policy architecture*: The proposed Block Transformer is designed for a very specific kind of task which limits its wider applicability. The authors “suppose the task is to relocate objects with the matched attributes into the goal position” (lines 242-243), which is not just treated as an example but determines the processing of state and goal slots. The block-wise cross-attention only allows matching objects to attend to each other which is a strong inductive bias that limits its expressivity. What if the task requires reasoning about the relationships of multiple goal objects? What if the task is to bring all the objects to the location of a single predefined object? Every change in the task/reward incurs a change to the architecture, which is not a desirable property. Judging by the results in Figure 11, this limited expressivity seems to not be beneficial even for the task it was designed for, resulting in lower sample efficiency compared to the EIT with the same representation.

Limitations of the approach should be discussed in the paper, possibly in the Conclusion section.

**Experiments**

*3D Block Slot Representation*: it is not clear how much of the improvement in the metrics you considered stem from the additional supervision versus the additional block structure. Experiments evaluating OSRT with the mask objective would help shed light on this matter.

*Goal-conditioned RL*: As I see it, the strong assumptions about the task embedded within the architecture are the main reason for the improved performance on the unseen test tasks. Since the matching mechanism is not learned, these tasks do not require much generalization in the learning sense, would the authors agree with this statement? Thus, although these findings are interesting, they could be considered narrow. I find the results in Figure 11 much stronger and I suggest they be moved to or at least discussed in the main text. The explicitly 3D-aware representations result in significant gains in sample efficiency compared to the 2D representation when paired with the EIT, which points to the fact that they facilitate better in-distribution generalization of the RL policy. Please see my suggestion under **Questions**.

*Misc*:
- How many seeds were used for the experiments?
- How many objects are present in the RL environments? It is not explicitly stated in the paper.
- Are the OOD colors also unseen during the representation model training or just the RL agent?
- I suggest highlighting results that are within a STD from the best performing method in Table 2.

**Questions:**

**Suggestion**

In my eyes, the main contribution of the paper is the 3D-aware object-centric representation and its applicability for training RL agents that generalize. This view is based on the experimental results, but is not highlighted by the structure and sentiment of the paper (other than the title).
The following suggestion is structural and semantic, and is of course left to the discretion of the authors, i.e., it will not affect my recommendation for acceptance/rejection:

Center the contribution around the 3D block-slot representation, and present the block transformer policy as an *example* of how this representation can facilitate design of task-specific architectures (although I do not think policy architectures should be task-specific). I would move the results in Figure 11 to the main text and discuss how with a fixed policy architecture:
1. 3D-aware representations can improve in-distribution generalization: evidence to that is the sample efficiency of OSRT and your representation compared to DLP on ID.
2. 3D-aware representations with factored attribute structure (i.e., yours) can improve attribute-level compositional generalization: evidence to that is the performance of yours compared to OSRT in CG and CG (same color).
3. Maybe discuss why OSRT performs better on OOD generalization with the same architecture.

**Questions**

- How much does the representation rely on knowing the amount of objects/slots in advance?
- How stable is training the block slot representation in terms of disentanglement of objects to distinct slots and factors to distinct blocks? Does the desired factorization emerge consistently across training runs?
- There are no details about the attribute matching mechanism in the block Transformer. Can the authors provide further details on how this is done?
- Are the authors planning to release code or take other measures for reproducibility purposes?
- A major benefit of the 3D aware representation in my opinion is that it may be view-agnostic. Have you considered training the policy when randomizing the viewpoints (maybe with some restrictions on their directions such that they will capture sufficiently different angles of the scene)? Demonstrating view-generalization capabilities can have a very large impact on the robotics community since vision-based control methods are largely not robust even to slight variations in camera angles to my impression.

---

> ### Author Response · Authors · 2025-11-20
>
> **W1. Abstract**
>
> Thank you for pointing this out. We agree that the original phrasing caused ambiguity. Our abstract wording may indeed have been unclear. We therefore replaced “shared concept memory” with the more standard “prototypical representation learning” and clarified “stably identify and track 3D trajectories” to “stably identify proxies of 3D positions.” These revisions remove potential ambiguity for general readers.
>
>
> **W2. Introduction**
>
> We appreciate the reviewer’s comment and agree that the two challenges were not sufficiently supported in the original text. We have revised the introduction to clearly articulate why insufficient 3D-awareness and imprecise object description arise, and we now cite prior work showing these limitations in existing 2D or single-view slot-attention models. The updated paragraph explains the concrete causes—missing multi-view reasoning, UV-grid decoding, and ambiguous unsupervised slot assignment—and grounds them in relevant literature. These additions clarify that our claims are evidence-based rather than speculative.
>
> **W3. Assumptions and limitations**
>
> **Agent and background masks**
>
> We agree that our method is not fully unsupervised, as our implementation uses GT agent and background masks. However, the method does not require access to these GT masks: agent masks can be rendered from the robot’s URDF, joint states, and calibrated cameras from hand-eye calibration, and coarse BG masks can be obtained from DINO’s foreground estimates. Examples of these approximate masks are shown in Figure 15. These suboptimal masks are sufficient for training, and our experiments show no performance degradation without high-fidelity GT masks, as reported in Table 14. Furthermore, using object-centric features from large foundation models would not resolve this issue, as such representations are not attribute-disentangled and are typically too heavy for per-frame inference in robotic control.
>
> **Identifying the meaning of latent block attributes**
>
> We appreciate this point. Our method yields human-interpretable latent variables through 3D block-slot learning, though the original version required offline analysis to identify which block encodes which attributes. Our prior diagnostics—K-means clustering and block manipulation tests—already showed that blocks align with meaningful object attributes without any labels (Appendix B.1). In response to the reviewer’s concern, we now describe an automatic procedure: by measuring block-wise temporal variance, we separate dynamic from static blocks and use only the static ones for Hungarian matching. This removes the need for manual identification and confirms that our block-slots contain reliably structured attribute information.
>
>
> **Strong assumption about a specific task embedding w/i policy architecture**
>
> Thank you for this thoughtful assessment. Our current policy is designed for object-wise goal-conditioned RL, and we agree it is less suited for modeling complex multi-object interactions. While some interactions are captured through self-attention and the final MLP aggregator of block transformer policy, these components are limited in expressiveness. More hierarchical goal-selection mechanisms—such as those in SMORL [1] or SRICS [2]—offer a promising direction for richer inter-object reasoning. We view this as an important avenue for strengthening our policy architecture in future work.
>
> [1] Zadaianchuk, A., Seitzer, M., & Martius, G. (2020). Self-supervised visual reinforcement learning with object-centric representations. arXiv preprint arXiv:2011.14381.
>
> [2] Zadaianchuk, A., Martius, G., & Yang, F. (2022, January). Self-supervised reinforcement learning with independently controllable subgoals. In Conference on Robot Learning (pp. 384-394). PMLR.
>
> **W4. Experiments**
>
> **3D Block-Slot Representation**
>
> We appreciate the reviewer's careful comparison. We also evaluated a variant where all slots use vanilla slot-attention (i.e., OSRT with our auxiliary mask loss; Appendix B.4, Table 9). This model performs worse in PSNR, FG-ARI, and DCI. The drop occurs because our approach relies on structured updates—background and agent slots use mask-guided updates distinct from object slots—whereas OSRT applies a single uniform update to all slots, preventing robust decomposition and disentanglement.

---

> ### Author Response · Authors · 2025-11-21
>
> **Goal-conditioned RL**
>
> Thank you for bridging attention to this point. Hungarian matching is a standard and widely used mechanism [1][2] in object-centric learning for maintaining permutation-consistent slots. Our use of it does not encode task-specific assumptions but follows this established practice. While the matching itself is not learned, the policy must still generalize over novel object configurations and viewpoints through the learned 3D block-slot representations, so the improvement is not attributable to matching alone.
>
> [1] Veerapaneni, R., Kossen, T., Kipf, T., van der Pol, E., Scholkopf, B., & van Steenkiste, S. (2020). OP3: Object-Oriented Prediction and Planning. Conference on Robot Learning (CoRL).
>
> [2] Zadaianchuk, A., Seitzer, M., & Martius, G. (2020). Self-supervised visual reinforcement learning with object-centric representations. arXiv preprint arXiv:2011.14381.
>
> **Suggestion**
>
> Thank you for the insightful suggestion. We agree that highlighting the 3D block-slot representation more centrally would strengthen the paper and will reflect this in the final version. As the reviewer noted, the view-generalization results clearly demonstrate the benefit of our 3D-aware representation, which is why we chose to emphasize them.
>
> **Q1. How many seeds were used for the experiments?**
>
> We used three random seeds for each experiment.
>
> **Q2. How many objects are present in the RL environments? It is not explicitly stated in the paper.**
>
> Our IsaacGym environments consist of two random objects.
>
> **Q3. Are the OOD colors also unseen during the representation model training or just the RL agent?**
>
> Our OOD color setting pretrains all vision models on IsaacGym3D, which includes OOD colors, but withholds those colors during RL so the policy never encounters them during training. The same applies to the baselines (DLP and OSRT). The strong OOD performance in Table 2 shows that our policy leverages attribute-aware reasoning rather than relying solely on object-centric features, enabling better handling of unseen attribute variations.
>
> **Q4. How much does the representation rely on knowing the amount of objects/slots in advance?**
>
> In our setup, both vanilla slot-attention and block-slot attention use a fixed slot count, following standard slot-based architectures. We acknowledge recent work exploring dynamic slot allocation [1] and agree that such techniques are complementary to our contributions. Incorporating such techniques is orthogonal to our main contribution and could enable the policy to better adapt to varying numbers of objects in robotic scenes. We view integrating dynamic slot allocation as a promising future direction that can further strengthen the flexibility of our approach.
>
> [1] Fan, Z., Zhang, Y., Zeng, A., & Xu, D. (2024). Adaptive Slot Attention: Object Discovery with Dynamic Slot Number.arXiv preprint arXiv:2406.09196.
>
> **Q5. How stable is training the block slot representation in terms of disentanglement of objects to distinct slots and factors to distinct blocks? Does the desired factorization emerge consistently across training runs?**
>
> As shown in Appendix B.2 and Figure 11, the desired factorization emerges reliably across runs. Training proceeds through two stable transitions—first learning the agent slot, then the object slots—after which reconstruction and mask losses converge smoothly. Block-wise attribute separation appears shortly after successful object reconstruction and becomes increasingly disentangled, yielding a stable block-wise structure across runs.
>
> **Q6. There are no details about the attribute matching mechanism in the block Transformer. Can the authors provide further details on how this is done?**
>
> Our block-wise cross-attention matches object slots between state and goal using only static attribute blocks, avoiding reliance on dynamic attributes like position. This is enabled by the pre-trained 3D block-slots, which yield semantically meaningful block representations (validated via block manipulation and K-means). After matching, cross-attention operates on fixed block indices, allowing consistent attribute-wise comparisons across objects.
>
> **Q7. Are the authors planning to release code or take other measures for reproducibility purposes?**
>
> We will release the full implementation publicly available soon.
>
> **Q8. View generalization capabilities**
>
> Thank you for this insightful suggestion. We agree that 3D-aware representations naturally support strong view-generalization. Because our encoder does not rely on any fixed camera angle, it retains task-relevant information across viewpoint changes. As shown in Appendix B.2, even a single-view input yields PSNR, FG-ARI, and DCI scores from one observation. Likewise, a policy trained with three views maintains its performance when given only a single view at test time, confirming that the block transformer policy relies on view-agnostic attribute representations for both RL and vision tasks as shown in Table 3.

---

> > ### Comment · Reviewer_xPHR · 2025-11-21
> >
> > Thank you for a very thorough response that has addressed most of my concerns.
> > The additional results on ablating the structure of the slot-attention modules, viewpoint generalization and use of approximate masks strengthen the paper’s contribution.
> >
> > Regarding the goal-conditioned RL point, I would like to reiterate that matching state and goal objects explicitly with the Hungarian algorithm and allowing only matched objects to attend to each-other in the cross-attention layers within the architecture is a *task-specific specific assumption*. See my initial point for examples where this inductive bias is not appropriate.
> > That said, I do not see this as the main contribution of the paper and thus will not further pursue this point.
> >
> > A discussion of the *limitations* of your approach would be beneficial as part of the conclusion section.
> >
> > Please explicitly mention in the paper:
> > 1. The use of the Hungarian algorithm for matching.
> > 2. The number of seeds used for evaluation.
> > 3. The number of objects in the GCRL experiments.
> >
> > Following your response and additional results, I believe this is a good paper and should be accepted. I have raised my score from 6 to 8 accordingly.

---

> > > ### Author Response · Authors · 2025-11-22
> > >
> > > We agree that directing cross-attention through Hungarian-matched objects has certain limitations and may not apply equally to all tasks. We have added a brief discussion of these limitations in the conclusion section and explicitly included details you requested in the paper.
> > >
> > > Thank you for your thoughtful and supportive comments. We appreciate the time you devoted to reviewing our work, and your insights greatly helped improve the clarity and quality of the paper.

---

### Official Review · Reviewer_BVmH · 2025-11-01

**Soundness:** 2
**Presentation:** 2
**Contribution:** 3
**Rating:** 6
**Confidence:** 4

**Summary:**

This paper proposes a 3D block-slot representation for vision-based reinforcement learning (RL), targeting improved compositionality, interpretability, and 3D-awareness in multi-object manipulation domains. The method integrates a multi-view transformer encoder with a novel block-slot attention mechanism, decomposing object representations into attribute-level blocks (e.g., color, shape, position) and separating active (objects), passive (background), and agent entities. The paper introduces a block transformer policy leveraging these structured representations for goal-conditioned RL, aiming to boost generalization in out-of-distribution and compositional scenarios. Experimental results are presented on Clevr3D and IsaacGym3D, highlighting gains in object decomposition, disentanglement, and RL performance compared to strong scene-centric and object-centric baselines.

**Strengths:**

1. **Explicit 3D Disentanglement**: The proposed 3D block-slot attention mechanism, as described in Section 2.1.3, provides structured disentanglement of object attributes into interpretable blocks (e.g., shape, color, size, position), enabling clear alignment with compositionality goals. This is well illustrated in Figure 3 and further analyzed in Figure 8 and Figure 9, which show that certain blocks consistently encode specific semantics (e.g., shape, color).
2. **Separation of Agents, Foreground, Background**: The method distinguishes between active (object), passive (background), and agent entities, as detailed in Section 2.1.2, with dedicated auxiliary objectives enforcing slot assignment and improving downstream policy stability. Figure 6 demonstrates visually superior decomposition versus OSRT, where the agent and background are reliably isolated.
3. **Goal-Conditioned Block Transformer Policy**: The block transformer policy (Section 2.2, Figure 2 and Figure 5) leverages block-level cross-attention for precise goal conditioning, explicitly matching object attributes in current and target scenes. Table 2 quantitatively supports the claim that this block-level policy bolsters generalization, especially in compositional and OOD tasks.

**Weaknesses:**

1. **Limited Diversity of Environments**
   The evaluation is restricted to Clevr3D and Isaac Gym 3D environments. Expanding the experiments to include additional benchmarks would provide stronger empirical evidence and reinforce the paper’s claims of effectiveness.

2. **Incomplete Baseline Comparisons**
   The work compares performance only against earlier object-centric models such as DLPv2 and OSRT. However, more recent and effective object-centric world models [1], [2] should also be considered. Even though these operate in 2D, their inclusion would clarify whether the 3D extension genuinely enhances object-centric policy quality.
   Additionally, comparisons with non-object-centric 3D baselines [3] and visual 2D non-object-centric RL approaches [4], [5] would help assess whether object-centric representations offer advantages for policy learning.

3. **Insufficient Ablation and Sensitivity Analysis**
   The paper introduces several architectural and training design choices that are not thoroughly examined.
   - The choice to apply block-slot attention only to foreground slots (Section 2.1.3), while using vanilla slot attention for agent and background, is motivated by intuition but lacks empirical validation. A comparison among different configurations—such as applying block-slot attention to all slots, a mixed setup, or using only vanilla slot attention—would strengthen the justification. Similarly, the auxiliary mask losses in Equation 4 are not ablated.
   - Key hyperparameters such as the number of slots, blocks (Table 3), and prototypes are chosen without detailed diagnostic analysis. The absence of sensitivity studies makes it difficult to assess reproducibility, robustness, and generalization of the proposed method.


[1] Ferraro et al. “FOCUS: Object-Centric World Models for Robotics Manipulation.” 2023. [https://arxiv.org/abs/2307.02427](https://arxiv.org/abs/2307.02427)


[2] Mosbach et al. “SOLD: Slot Object-Centric Latent Dynamics Models for Relational Manipulation Learning from Pixels.” 2024. [https://arxiv.org/abs/2410.08822](https://arxiv.org/abs/2410.08822)

[3] Shim et al. “SNeRL: Semantic-aware Neural Radiance Fields for Reinforcement Learning.” 2023. [https://proceedings.mlr.press/v202/shim23a.html](https://proceedings.mlr.press/v202/shim23a.html)

[4] Hafner et al. “Mastering Diverse Domains through World Models.” 2023. [https://arxiv.org/abs/2301.04104](https://arxiv.org/abs/2301.04104)

[5] Zhou et al. “DINO-WM: World Models on Pre-trained Visual Features enable Zero-shot Planning.” 2024. [https://arxiv.org/abs/2411.04983](https://arxiv.org/abs/2411.04983)

**Questions:**

1. See weaknesses section
2. Could you share insights or figures analyzing the impact of number of slots, blocks and prototypes on both decomposition and RL quality?
3.  It is unclear whether the permutation of slots between the current and goal images influences the policy’s performance. Does the model explicitly handle slot alignment or permutation, or could mismatched slot ordering negatively impact policy learning?

---

> ### Author Response · Authors · 2025-11-20
>
> **W1. Limited Diversity of Environments**
>
> Thank you for raising this point. Our goal is to evaluate how our 3D-aware object-centric representation improves policy learning and view-generalization, rather than benchmark across many environments. While the number of environments is limited, our setup includes strong structure variations—randomized viewpoints, object permutations, occlusions, and explicit background-agent decomposition—which provide strong stress tests for the core representation module. We agree that expanding to more environments would further strengthen the study and plan to explore this in future work.
>
> **W2. Incomplete Baseline Comparisons**
>
> We appreciate the reviewer’s careful consideration of the baseline choices. World-model approaches aim to learn latent dynamics and perform imagination or planning, whereas our work focuses on a model-free goal-conditioned policy trained purely on a pre-trained 3D-aware object-centric encoder. Because these methods differ fundamentally in purpose, architecture, and required redesign for multi-view control, they are not directly comparable within our representation-centric scope.
>
> Prior work has already demonstrated the generalization benefits of 2D object-centric over 2D non-object-centric representations [6][7], and we complement this by implementing SNeRL [3] as a meaningful 3D non-object-centric baseline. However, SNeRL did not achieve reliable novel-view synthesis in our IsaacGym3D dataset due to challenges posed by multi-object randomness, frequent occlusions, and highly dynamic robot configurations. This observation suggests that a more expressive encoder is needed in such settings: compared to SNeRL’s architecture, our transformer-based multi-view encoder is better suited for learning consistent representations across diverse object configurations and viewpoints. Empirical results supporting this observation are provided in Appendix A.2 and Figure 6.
>
> [1] Ferraro, S., Walsman, A., Xu, D., & Zeng, A. (2023). FOCUS: Object-Centric World Models for Robotics Manipulation. arXiv preprint arXiv:2307.02427.
>
> [2] Mosbach, M., Pirk, S., & Xu, D. (2024). SOLD: Slot Object-Centric Latent Dynamics Models for Relational Manipulation Learning from Pixels. arXiv preprint arXiv:2410.08822.
>
> [3] Shim, J., Kim, J., Kim, H., & Lee, K. M. (2023). SNeRL: Semantic-Aware Neural Radiance Fields for Reinforcement Learning. In Proceedings of the 40th International Conference on Machine Learning (ICML), PMLR.
>
> [4] Hafner, D., Pasukonis, J., & Lillicrap, T. (2023). Mastering Diverse Domains through World Models. arXiv preprint arXiv:2301.04104.
>
> [5] Zhou, J., Hafner, D., & Pathak, D. (2024). DINO-WM: World Models on Pre-trained Visual Features Enable Zero-shot Planning. arXiv preprint arXiv:2411.04983.
>
> [6] Yoon, A., Park, J., Lee, S., & Kim, J. (2023). An Investigation into Pre-Training Object-Centric Representations for Reinforcement Learning. arXiv preprint arXiv:2302.04419.
>
> [7] Kapl, T., Balloch, J., Hadsell, R., & van Steenkiste, S. (2025). Object-Centric Representations Generalize Better Compositionally with Less Compute. In International Conference on Learning Representations (ICLR).
>
> **W3. Insufficient Ablation and Sensitivity Analysis**
>
> Thank you for pointing out the need for deeper analysis. As the reviewer suggested, we have added the following results to clarify the behavior of each design choice as well as the effect of key hyperparameters, and we report them in Appendix B.4.
>
> **Mixture structure of slot-attention**
>
> Our mixed architecture is essential for stable background, agent, and object decomposition and robust 3D block-wise attribute learning. With vanilla slot-attention alone, background and agent slots can be guided by the auxiliary mask loss, but object slots cannot acquire attribute-wise block latents—preventing effective use of the block transformer policy that underlies our generalization results. Using only block-slot attention, however, forces background and agent slots to learn object-style blocks, breaking both decomposition and attribute modeling, as shown in Figure 14. The mixed design resolves these issues by applying vanilla slot-attention to background and agent slots and block-slot attention to object slots.
>
> **Ablation study of auxiliary mask loss scale**
>
> Our ablation varying $\lambda_{bg}$ and $\lambda_{ag}$ (0.1, 0.5, 1.0) shows that large scales increase FG-ARI but reduce PSNR and block-wise disentanglement. When the mask loss dominates the reconstruction loss, the model overemphasizes mask-based decomposition at the cost of reconstruction fidelity and attribute consistency. Nonetheless, the degradation is modest, and all settings still yield stable background, object and agent separation—crucial for downstream block-slot reasoning. Quantitative results are provided in Table 13.

---

> ### Author Response · Authors · 2025-11-20
>
> **Number of blocks**
>
> We ablate the number of blocks (4, 8, 16) while keeping the slot dimension fixed and observe that more blocks consistently improve disentanglement. Smaller blocks limit representational capacity, encouraging attributes to separate across blocks rather than mix. With fewer, larger blocks, multiple attributes can reside in the same block, leading to entangled representations. See Table 10 for detailed results.
>
> **Number of prototypes**
>
> We also varied the number of prototypes (8, 16, 32). With only 8 prototypes, the model can still decompose objects and synthesize novel views, but block-wise attribute separation breaks down due to limited expressive capacity. Increasing prototypes consistently improves PSNR, FG-ARI and DCI, indicating that richer prototype sets better capture subtle variations in color, shape, size, and position. Thus, prototype granularity is crucial for strong block-wise disentanglement. Corresponding quantitative results are shown in Table 11.
>
> **Number of slots**
>
> In our model, the slot count is fixed and purposely chosen to exceed the number of objects, causing extra slots to naturally remain empty—an established property of slot-attention models. In practice, increasing or decreasing the slot count does not influence decomposition as long as it exceeds the true number of objects, making slot-count ablation uniformative. While recent work explores dynamically adjusting the number of  slots [1], such mechanisms are orthogonal to our contribution. We agree that integrating dynamic slot allocation is an interesting future direction for improving flexibility.
>
> [1] Fan, Z., Zhang, Y., Zeng, A., & Xu, D. (2024). Adaptive Slot Attention: Object Discovery with Dynamic Slot Number. arXiv preprint arXiv:2406.09196.
>
> **Q2. Could you share insights or figures analyzing the impact of number of slots, blocks and prototypes on both decomposition and RL quality?**
>
> Thank you for raising this point. Our decomposition ablations already reveal clear trends: increasing the number of blocks or prototypes consistently improves attribute disentanglement, which directly strengthens the representations consumed by the policy. In contrast, the slot count has negligible impact as long as it exceeds the number of objects, since unused slots naturally remain empty under standard slot-attention dynamics.
> These findings suggest that RL policies mainly benefit from improved disentanglement, which yields clearer cross-attention signals and reduces static attribute ambiguities in object matching. The corresponding RL results are reported in Table 12.
>
> **Q3. It is unclear whether the permutation of slots between the current and goal images influences the policy’s performance. Does the model explicitly handle slot alignment or permutation, or could mismatched slot ordering negatively impact policy learning?**
>
> Thank you for raising this question. Our block transformer includes a Hungarian-matching step to align object slots before performing block-wise cross-attention. Because object slots are permutation-invariant, index alignment between the current state and the goal state is not guaranteed. We therefore match slots using only static attribute blocks—excluding position—to obtain a stable object-wise correspondence. This alignment enables reliable cross-attention and ensures the policy consistently attends to the correct object-specific blocks.

---

> ### Comment · Reviewer_BVmH · 2025-11-26
>
> Thank you for your response! While I understand that there are not many goal-conditioned, object-centric baselines, I still believe it would be valuable to compare this model against at least a simple CNN or DINO encoder (using the same policy) to demonstrate that the 3D modality actually improves slot quality. That said, the majority of my concerns have been addressed. I think it’s great work, and I am willing to raise my score.

---

> > ### Author Response · Authors · 2025-11-27
> >
> > We agree that including a comparison with a CNN or DINO encoder would better highlight the benefits of the 3D modality, and we will incorporate these experiments in the final version.
> >
> > Thank you for your thoughtful and supportive feedback. We truly appreciate the effort you put into evaluating our work, and your constructive suggestions have been instrumental in strengthening both the clarity and overall quality of the paper.

---

### Official Review · Reviewer_PdX4 · 2025-11-01

**Soundness:** 2
**Presentation:** 3
**Contribution:** 3
**Rating:** 4
**Confidence:** 2

**Summary:**

The paper proposes a 3-D block-slot encoder–decoder architecture that fuses a multi-view Object Scene Representation Transformer (OSRT) with a block-slot decomposition module inspired by SysBinder. Each scene is encoded into a set of latent “blocks,” each corresponding to an object slot that factorises attributes such as shape, colour, size, and position. Two auxiliary supervision losses constrain fixed background and agent slots.
On top of this representation, a Block Transformer (BT) policy performs block-wise cross-attention between current and goal states for goal-conditioned manipulation.

**Strengths:**

1. The paper correctly identifies that prior slot-based encoders struggle with 3-D occlusion and pose entanglement, and that OSRT-like models lack attribute factorization. The system architecture is easy to follow, and ablations are reasonably thorough.

2. The combination of OSRT’s volumetric scene reasoning with SysBinder-style factorized slots is a clever, pragmatic design. The explicit background and agent slots make the representation interpretable and stable across timesteps.

3.  On both Clevr3D and IsaacGym3D, the model shows clear quantitative gains in decomposition (FG-ARI and D/C/I) and control success. The Block-Transformer (BT) policy demonstrably improves compositional generalization, especially in out-of-distribution color/shape settings.

**Weaknesses:**

1. The method’s conceptual innovation is modest. The encoder is largely a direct hybrid of OSRT (multi-view scene transformer for volumetric representation) and SysBinder (slot factorisation via attribute prototypes). The Block Transformer (BT) is a small architectural tweak of EIT, with cross-attention operating at the block level instead of the pixel/feature level. This makes the contribution primarily engineering integration rather than algorithmic advancement. The claimed novelty of “3D-aware block-slot” reduces to combining volumetric rendering with slot decomposition.

2. A major conceptual gap is the reliance on explicit segmentation masks for two dedicated slots: Eq. (2) defines auxiliary losses requiring binary masks for the background and agent (manipulator). Without such supervision, it’s unclear whether the model can autonomously discover these roles. This violates the unsupervised or weakly-supervised assumption common in object-centric learning, where slots are meant to emerge purely from reconstruction signals. In real-world robotics, such masks are rarely available. In unlabeled multi-object scenes, the model might not allocate the correct number of slots or separate the manipulator from movable objects.

3. Several notational and definitional issues appear: In Eq. (1), the attention-weight mask loss compares per-ray weights
𝑤 with 2D masks 𝑚; the mapping from ray to pixel is unspecified, making the loss potentially ill-posed.

In Eq. (3), λ_bg and λ_ag are introduced but not analyzed, their effect on disentanglement or convergence is unknown.
The model description doesn’t clarify how 3D latent “blocks” correspond to volumetric regions or voxels; thus, the claim of 3D-awareness is more architectural than mathematical.

4. Does not currently release its implementation or code repository.

**Questions:**

1. How does performance degrade if only a single camera is available at test time? Can the model lift monocular views to 3-D reliably?

2. Please report inference latency per frame and total compute (encoder + BT) versus DLPv2 and OSRT baselines.

3. If background/agent masks are unavailable, does the model still allocate them automatically or does performance collapse?

---

> ### Author Response · Authors · 2025-11-20
>
> **W1. The method’s conceptual innovation is modest.**
>
> Thank you for raising this concern. Our hybrid perception architecture provides a novel and effective solution for robust scene decomposition in realistic robotic environments. Specifically, we are the first to unify vanilla slot-attention and block-slot attention within a single framework, supported by an auxiliary loss. This design enables a structured decomposition that neither OSRT nor SysBinder—nor their naive combination—can achieve. Experiments further confirm its effectiveness, showing that object slots learn block-wise attributes while background and agent slots remain cleanly and stably separated.
>
> **W2. A major conceptual gap is the reliance on explicit segmentation masks for two dedicated slots.**
>
> Thank you for pointing out this concern. Our method does not rely on high-fidelity ground-truth masks, and our experiments show that its performance remains stable even without them (see Appendix B.4 and Table 14 for quantitative results). Although our implementation uses GT agent and background masks, the model can operate with approximate masks obtained without supervision. The agent mask can be rendered directly from the robot’s URDF, joint states, and calibrated camera pose, while coarse BG masks are derived from DINO-based foreground estimation (see Figure 15 for examples). Together, these sub-optimal masks are sufficient for training and preserve the effectiveness of our approach.
>
> **W3. Several notational and definitional issues appear: In Eq. (1), the attention-weight mask loss compares per-ray weights 𝑤 with 2D masks 𝑚; the mapping from ray to pixel is unspecified, making the loss potentially ill-posed.**
>
> We appreciate the opportunity to clarify this formulation. Our auxiliary loss supervises background and agent slots through their pixel-level contributions rather than an $\mathcal{L}_2$ loss on slot weights. The revised formulation aligns each slot’s attention-weighted predicted pixels with the corresponding mask-weighted ground-truth pixels. This correction reflects the actual implementation and clarifies how the slots are guided toward their semantic roles during training.
>
> **W3. In Eq. (3), λ_bg and λ_ag are introduced but not analyzed, their effect on disentanglement or convergence is unknown.**
>
> We agree that this clarification is important. Our model consistently embeds the background and agent into their designated slots once the auxiliary mask loss is applied, regardless of the specific loss scale. To support this, we analyze the auxiliary mask loss in Appendix B.4 by varying $\lambda_{bg}$ and $\lambda_{ag}$ across 0.1, 0.5, and 1.0 and reporting PSNR, FG-ARI, and DCI (see Table 13 for full results). While setting them to 1.0 slightly harms reconstruction and disentanglement by overpowering the reconstruction loss, the degradation is minimal. Overall, the mask loss reliably enables stable BG/AG separation and preserves the model’s ability to obtain attribute-wise 3D blocks.
>
> **W3. The model description doesn’t clarify how 3D latent “blocks” correspond to volumetric regions or voxels; thus, the claim of 3D-awareness is more architectural than mathematical.**
>
> Thank you for raising this interpretability question. Our 3D block-slot representation conveys genuine 3D information even without explicit voxels. Multi-view consistency allows each object slot to recover 3D geometry in the same way that light-field representations infer depth through stereo cues [1]. The block decomposition then disentangles object-specific 3D attributes, functioning as implicit volumetric parts rather than explicit grids. Thus, the model’s 3D-awareness arises from learned multi-view geometry, not merely from architectural design.
>
> [1] Wanner, S., & Goldluecke, B. (2012). Globally consistent depth labeling of 4D light fields. In Proceedings of the IEEE Conference on Computer Vision and Pattern Recognition (CVPR), 41–48.
>
>
> **W4. Code implementation**
>
>
>  Thank you for noting this. We will release the full implementation publicly avaialbe soon.

---

> ### Author Response · Authors · 2025-11-20
>
> **Q1. How does performance degrade if only a single camera is available at test time? Can the model lift monocular views to 3-D reliably?**
>
> We appreciate the reviewer’s interest in this setting. Our 3D-aware block-slot representation reliably lifts even a single monocular view into a consistent 3D attribute space, enabling robust performance at test time. As shown in Table 3, the policy trained on three fixed views generalizes well to randomized single-view and multi-view conditions, with only a minor drop due to unavoidable occlusions. Consistently, Appendix B.2 (Table 7) shows that the vision model itself achieves nearly identical novel-view synthesis and decomposition performance under single-view inference. These results confirm that one view is already sufficient for meaningful 3D reasoning, object decomposition, and attribute-based control.
>
> **Q2. Please report inference latency per frame and total compute (encoder + BT) versus DLPv2 and OSRT baselines.**
>
> Thank you for the request. We report the inference latency per frame (encoder + policy) and the total wall-clock training time for baselines. As shown below, our model has moderately higher per-frame latency due to block-slot attention, while remaining within a practical range. The total compute is comparable to other multi-view baselines despite the increased representational capacity.
>
> | **Method**      | **Inference latency per frame** | **Total training time** | **Steps** |
> |-----------------|----------------------------------|--------------------------|-----------|
> | DLPv2 w/ EIT     | 1.691 ms                         | 2d 13h 28m               | 6.04M     |
> | OSRT w/ EIT      | 1.630 ms                         | 2d 3h 47m                | 6.04M     |
> | Ours w/ BT       | 2.930 ms                         | 2d 14h 50m               | 6.04M     |
>
>
> **Q3. If background/agent masks are unavailable, does the model still allocate them automatically or does performance collapse?**
>
> Thank you for highlighting this scenario. Our model requires background and agent masks; without them, all slots become permutation-invariant and no longer bind to their designated roles. However, obtaining suboptimal masks is always feasible: robot URDF, joint states, and calibrated camera poses allow us to render agent masks, and DINO reliably provides coarse background masks. These masks are sufficient for the auxiliary loss to enforce consistent slot indexing, letting the model stably separate background and agent and learn object attributes.

---

### Author Response · Authors · 2025-11-20

**Common Response: Additional Experiments and Clarifications**

We thank all reviewers for the constructive feedback. We performed additional experiments and expanded our analysis to more clearly demonstrate the robustness and novelty of our approach.

**1. Robustness of the Auxiliary Mask Loss.**

Our experiments show that the auxiliary mask loss remains stable across a range of weighting scales and does not require high-fidelity ground-truth masks. Suboptimal masks derived automatically from robot rendering and DINO-based foreground cues lead to nearly identical performance, indicating that the loss reliably enforces background and agent separation even under imperfect and unlabeled mask signals.

**2. Strong 3D-Aware and View-Agnostic Representations.**

We added experiments confirming that our 3D block-slot attention reliably infers 3D structure and disentangled object attributes—even from a single image—while maintaining performance across novel viewpoints and varying numbers of input views. These results highlight a core novelty of our representation: unlike 2D object-centric encoders, it yields view-agnostic attribute slots that support robust policy generalization.

**3. Comprehensive Ablations on Block-Slot Design.**
We evaluated the sensitivity of our model to the number of blocks and prototypes. More blocks or prototypes improve block-wise attribute disentanglement, confirming that our representation’s structure is responsible for the observed generalization benefits. These ablations collectively demonstrate that the 3D block-slot representation’s architecture is principled and stable.


Overall, these additions strengthen our claim that 3D block-slot representations produce semantically meaningful, 3D-aware, and view-agnostic representations that directly benefit downstream policy learning. Moreover, our block transformer policy leverages these block-wise representations effectively, enabling the strong generalization performance observed across all evaluation settings.

---

### Comment · Area_Chair_j2ue · 2025-11-24

Dear Reviewers,

The authors have responded to your reviews. Please review and respond to their comments who have not yet done so.

Best, Your AC

---

### Meta-Review · Area_Chair_RzQ1 · 2025-12-26

**Summary:**

An object centric representation that is 3D is proposed, and evaluated on goal conditioned RL. A main advantage of this approach compared to the state of the art is that it can handle camera viewpoint changes. Reviewers raised concerns about the novelty of the approach, the presentation of the paper, and some technical questions about baselines/ablations. The author rebuttal addressed most concerns.
While this work is relatively incremental in novelty, it does address an important shortcoming in the state of the art. Please take the reviewer suggestions into account when revising your paper.

**Reviewer Concerns:**

above

**Reviewer Scores:**

4,6,6->8

---

### Decision · Program_Chairs · 2026-01-26

Accept (Poster)